# A novel cascade allows *Metarhizium robertsii* to distinguish cuticle and hemocoel microenvironments during infection of insects

**Xing Zhang**, **Yamin Meng, Yizhou Huang, Dan Zhang, Weiguo Fang** *

MOE Key Laboratory of Biosystems Homeostasis & Protection, Institute of Microbiology, College of Life Science, Zhejiang University, Hangzhou, China

* wfang1@zju.edu.cn

**Data Availability Statement:** All relevant data are within the paper and its Supporting Information files. The GenBank accession number for all of the

## Abstract

Pathogenic fungi precisely respond to dynamic microenvironments during infection, but the underlying mechanisms are not well understood. The insect pathogenic fungus *Metarhizium robertsii* is a representative fungus in which to study broad themes of fungal pathogenicity as it resembles some major plant and mammalian pathogenic fungi in its pathogenesis. Here we report on a novel cascade that regulates response of *M. robertsii* to 2 distinct microenvironments during its pathogenesis. On the insect cuticle, the transcription factor COH2 activates expression of cuticle penetration genes. In the hemocoel, the protein COH1 is expressed due to the reduction in epigenetic repression conferred by the histone deacetylase HDAC1 and the histone 3 acetyltransferase HAT1. COH1 interacts with COH2 to reduce COH2 stability, and this down-regulates cuticle penetration genes and up-regulates genes for hemocoel colonization. Our work significantly advances the insights into fungal pathogenicity in insects.

## Introduction

Fungal pathogens of insects, plants, and mammals usually encounter dynamic microenvironments during infection of their hosts, but the mechanisms for them to respond and adapt to microenvironments are not well understood [1]. The entomopathogenic and endophytic fungus *Metarhizium robertsii* has been used as a model to study fungal pathogenesis in insects [2]. Infection occurs when conidia adhere to the cuticle of a susceptible insect host and produce germ tubes that differentiate into infection structures called appressoria. The appressoria produce infection pegs, which penetrate the cuticle via a combination of mechanical pressure and cuticle-degrading enzymes. Once reaching the insect hemocoel, the fungus undergoes dimorphism from hyphae to yeast-like cells (i.e., blastospores), and the insect is killed by a combination of fungal growth and toxins. Finally, the fungus grown in the hemocoel reemerges from the dead insect. In this pathogenesis progression, *M. robertsii* encounters 2 different microenvironments: the insect cuticle and the insect hemocoel. The insect cuticle is the first barrier against fungal infection, and the mechanisms for *M. robertsii* to breach this barrier are similar to those of some plant pathogenic fungi such as *Magnaporthe oryzae* and *Colletotrichum*

RNA-Seq data obtained in this study is
PRJNA637940.

**Funding:** This work was funded by the National
Natural Science Foundation of China (http://www.
nsfc.gov.cn/), and the grant numbers are
31672078 and 31872021. W.F received these two
grants. The funders had no role in study design,
data collection and analysis, decision to publish, or
preparation of the manuscript.

**Competing interests:** The authors have declared
that no competing interests exist.

**Abbreviations:** ChIP-qPCR, chromatin
immunoprecipitation quantitative PCR; ChIP-Seq,
chromatin immunoprecipitation sequencing; Co-IP,
coimmunoprecipitation; DID, dimer interface
domain; EMSA, electrophoretic mobility shift
assay; NLS, nuclear localization signal; ORF, open
reading frame; PDA, potato dextrose agar; qPCR,
quantitative PCR; qRT-PCR, quantitative reverse
transcription PCR; RNA-Seq, RNA sequencing;
SDY, Sabouraud dextrose broth supplemented
with 1% yeast extract; WT, wild-type.

*lagenarium* [3,4], which all form appressorial infection structures. Pathogenicity factors such
as protein kinase A and hydrophobins are functionally conserved in the development of the
appressorium between *M. oryzae* and *M. robertsii* [5,6]. The pathogenesis of *M. robertsii* also
resembles that of mammalian pathogenic fungi in many aspects, such as the ability to evade
the host innate immune system that has been conserved between insects and mammals [7].
Therefore, *M. robertsii* can be used as a representative fungus to study broad themes of fungal
pathogenicity. In addition, *Metarhizium* species are being developed as environmentally
friendly alternatives or supplements to chemical insecticides in biocontrol programs for agri-
cultural pests and vectors of disease [8]. Detailed mechanistic knowledge of fungal pathogenic-
ity in insects is therefore needed for optimal mycoinsecticide development and improvement.

The mechanisms for cuticle penetration have been extensively investigated, and many cuti-
cle-degrading genes and important regulators that control cuticle penetration have been
reported [9]. Recently, some factors have also been found to play important roles in hemocoel
colonization, including the sterol carrier Mr-NPC2a, which is responsible for acquisition of
host sterols to maintain the integrity of the cell membrane of the fast-proliferating hyphal bod-
ies [10]. The collagen-like protein MCL1 and toxic secondary metabolites such as destruxins
facilitate the evasion of the host innate immune system [11,12]. The siderophore for iron
metabolism is also an important factor for hemocoel colonization [13]. RNA sequencing
(RNA-Seq) analysis in our recent work showed that many genes switch off during the micro-
environmental transition from the insect cuticle to the hemocoel, including cuticle-degrading
enzymes [9]. Some of these cuticle-degrading enzymes, such as Pr1 proteases and the metallo-
proteases MrMep1 and MrMep2, are important virulence factors [14,15], and their expression
needs to be down-regulated in the hemocoel, where they would otherwise activate insect
immunity [16]. Despite the extensive characterization of fungal functional genes for the infec-
tion of insects, the detailed regulatory mechanisms underlying the response of *M. robertsii* to
different microenvironments remain to be explored.

In this study, we discovered a novel regulatory cascade that controls the response of *M.
robertsii* to the 2 microenvironments in its pathogenesis. On the insect cuticle, the transcrip-
tion factor COH2 (colonization of hemocoel 2) activates the expression of genes involved in
cuticle penetration. Once the fungus enters the hemocoel, the regulatory protein COH1 (colo-
nization of hemocoel 1) is expressed, which physically contacts the transcription factor COH2
to reduce its stability, resulting in the inhibition of the expression of genes for cuticle penetra-
tion and the up-regulation of genes for hemocoel colonization. We further found that the
expression of COH1 in the hemocoel results from the reduction in epigenetic repression con-
ferred by the histone deacetylase HDAC1 and the histone acetyltransferase HAT1.

## Results

### Identification of the regulatory protein COH1

In our previous RNA-Seq analysis [9], we found that a gene (MAA_08820, designated as *Coh1*,
as it is involved in colonization of the hemocoel) was highly expressed during hemocoel colo-
nization. In this study, hemolymph collected from last instar *Galleria mellonella* larvae (used
for all assays in this study) was treated with an anticoagulant, and fungal growth in the result-
ing hemocyte-containing hemolymph (hereafter called hemolymph) was used as an approxi-
mation of hemocoel colonization, hereafter called surrogate hemocoel colonization.
Penetration of the cuticle was achieved by incubating the fungus on the cuticle of *G. mellonella*
larvae. Unless otherwise indicated, the insect cuticle used in this study was from *G. mellonella*
larvae. Saprophytic growth was achieved by growing the fungus in the liquid medium Sabour-
aud dextrose broth supplemented with 1% yeast extract (SDY) or on potato dextrose agar

(PDA) plates. Root colonization refers to the growth of *M. robertsii* on the roots of *Arabidopsis thaliana*. When quantitative reverse transcription PCR (qRT-PCR) analysis was conducted with RNA samples from cuticle penetration, saprophytic growth, and root colonization, the Cq values (the PCR cycle number at which a sample reaction curve intersects the threshold line) were approximately 38 cycles, and no PCR products were detected on the agarose gel (Fig 1A), suggesting that the transcription level of *Coh1* was extremely low or was not detectable. However, qRT-PCR analysis and subsequent detection of PCR products on agarose gel confirmed that the *Coh1* transcript was expressed during the surrogate hemocoel colonization (Fig 1A). The PCR product of *Coh1* was also detected with RNA prepared from live *G. mellonella* larvae infected by *M. robertsii*, suggesting that this gene was expressed in the real hemocoel of the insects. When RNA from the insect cadavers that were mummified with the mycelium was used, no PCR product was detected, indicating that *Coh1* was not expressed during the necrotrophic stage (Fig 1A).

*Coh1* is a single-copy gene with a 525-bp open reading frame (ORF) that encodes a protein containing 174 amino acid residues. COH1 contains an Ecl1 domain (PFAM12855) that was also identified in the ECL1 proteins of the saprophytic yeasts *Saccharomyces cerevisiae* and *Schizosaccharomyces pombe*; these proteins are considered regulators though their functions have not been characterized [17]. However, BLASTp analysis using the full-length protein sequence of COH1 as a query identified no significant similarity ($>e^{-5}$) between *M. robertsii* COH1 and the yeast ECL1 proteins. No putative signal peptide or nuclear localization signal (NLS) was predicted in COH1. In the chromosome of *M. robertsii*, *Coh1* was distant from its adjacent genes, with its ORF being 111,146 bp away from that of the upstream gene (MAA_10633) and 7,498 bp away from that of the downstream gene (MAA_08821).

## COH1 is involved in real hemocoel colonization

To investigate the biological functions of *Coh1*, we constructed a deletion mutant (Δ*Coh1*) that had the entire ORF deleted (S1A and S1B Fig). The genes and fungal strains used in this study are listed in Table 1. Southern blot analysis showed that there was only 1 copy of the herbicide-resistance gene *Bar* (the selection marker gene for fungal transformation) in 3 randomly selected isolates (Δ*Coh1*#1, Δ*Coh1*#2, and Δ*Coh1*#3) (S1C Fig), indicating that the insertion of the foreign DNA only resulted in deletion of *Coh1* via homologous recombination, and no ectopic integration occurred. The 3 independent isolates of Δ*Coh1* were subjected to the following phenotypic analysis, and no differences were observed between these isolates in all assays (see below). On the PDA plates, Δ*Coh1* was not different from the wild-type (WT) strain in colony growth, colony morphology, and conidial yield (S2A–S2C Fig).

The pathogenicity of *M. robertsii*, shown by the time taken for 50% of insects to die (LT$_{50}$), was assayed in *G. mellonella* larvae. Inoculations were conducted either by topically applying conidia onto the insect cuticle or by direct injection of conidia into the hemocoel (thus bypassing the cuticle). Via topical application, the LT$_{50}$ values of the 3 isolates of the mutant Δ*Coh1* were approximately 2-fold higher than that of the WT strain (Fig 1B). Via direct injection, the 3 isolates of Δ*Coh1* also had higher LT$_{50}$ values than the WT strain (S2D Fig). At 3 and 4 d after direct injection, the hemocoel of the insects injected with each of the 3 isolates of Δ*Coh1* contained significantly fewer ($P < 0.05$) hyphal bodies than the WT strain (S2E Fig). Quantitative PCR (qPCR) analysis further showed that the fungal burden in the dying insects infected by Δ*Coh1* was significantly lower than that of insects infected with the WT strain ($P < 0.05$) (S2F and S2G Fig). On a normally inductive milieu (i.e., the hydrophobic surfaces of plastic petri dishes) in the presence of low levels of nitrogenous nutrients, no difference in appressorium formation was found between the WT strain and the mutant Δ*Coh1* (S2H Fig). The

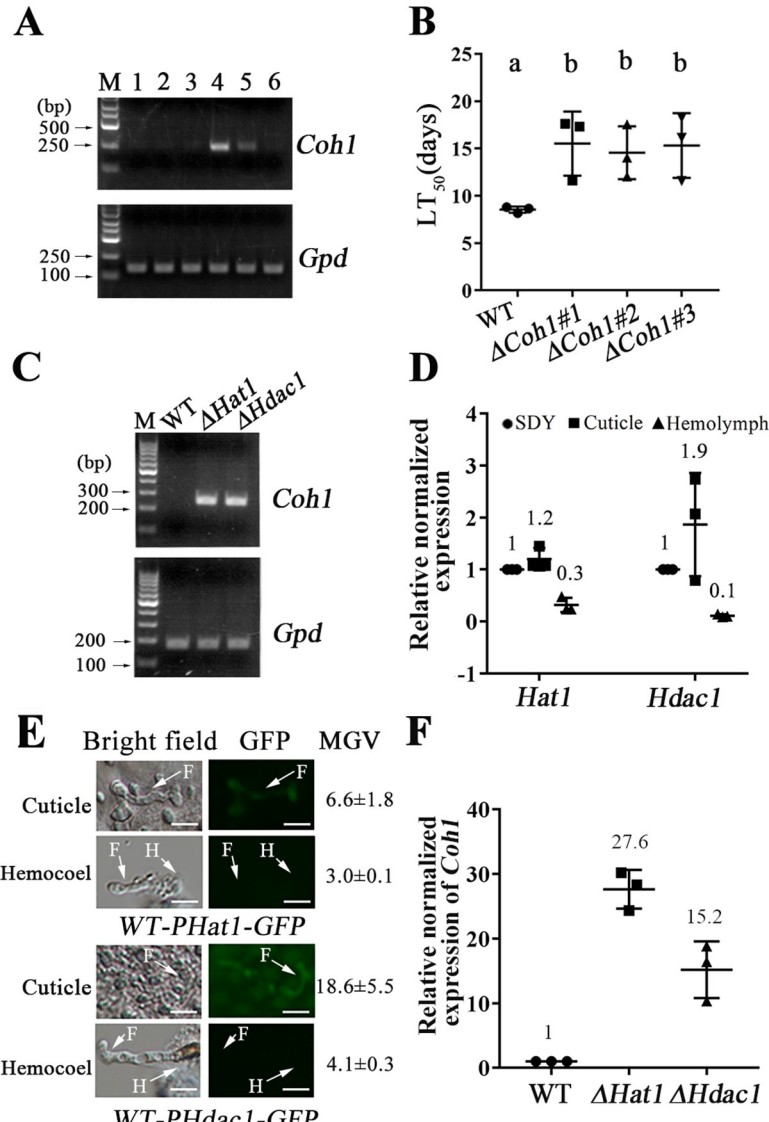

**Fig 1. Expression and regulation of the *Coh1* gene.** (A) Agarose gel electrophoresis of quantitative reverse transcription PCR (qRT-PCR) products of *Coh1*. 1: Saprophytic growth in Sabouraud dextrose broth supplemented with 1% yeast extract (SDY) medium; 2: cuticle penetration; 3: colonization of plant roots; 4: growth in the hemocyte-containing hemolymph; 5: live insects infected with *M. robertsii*; 6: insect cadavers mummified with *M. robertsii* mycelium; M: DNA ladder. Images are representative of 3 independent experiments. Upper panel: the gene *Coh1*; lower panel: the reference gene *Gpd* encoding glyceraldehyde 3-phosphate dehydrogenase. (B) $LT_{50}$ (time taken to kill 50% of insects) values when the insects were inoculated by topical application of conidia on the cuticle. Δ*Coh1#1*, Δ*Coh1#2*, and Δ*Coh1#3* are 3 independent isolates of the deletion mutant Δ*Coh1*. WT, wild type. The bioassays were repeated 3 times with 40 insects per repeat. Data are expressed as mean ± SE. Values with different letters are significantly different ($n = 3$, $P < 0.05$, Tukey's test in one-way ANOVA). (C) Agarose gel electrophoresis of qRT-PCR products of *Coh1* in WT strain and deletion mutants of the histone deacetylase gene *Hdac1* (Δ*Hdac1*) and the histone acetyltransferase gene *Hat1* (Δ*Hat1*) during saprophytic growth. Note: No qRT-PCR product was seen in the WT strain. (D) qRT-PCR analysis of the expression of *Hdac1 and Hat1* in the WT strain during the surrogate hemocoel colonization (Hemolymph) and cuticle penetration (Cuticle) relative to saprophytic growth (SDY). (E) GFP signal in the fungal cells on the cuticle (Cuticle) and in the real hemocoel (Hemocoel) of *G. mellonella* larvae. The strain with the *gfp* gene was driven by the promoter of *Hat1* (upper panels) or *Hdac1* (lower panels). F, fungal cells; H, hemocyte. Scale bar: 10 μm. Images are representative of 3 independent experiments. The mean gray value (MGV) shows the GFP fluorescence intensity in the fungal hyphae. (F) qRT-PCR analysis of *Coh1* expression in the WT strain and the mutants Δ*Hat1* and Δ*Hdac1* during the surrogate hemocoel colonization. All qRT-PCR experiments in this study were repeated 3 times. For qRT-PCR analyses in this figure, the values represent the fold-change of expression of a gene in treatment compared with expression in its respective control, which is set to 1. The data underlying all the graphs shown in this figure can be found in S1 Data.

**Table 1. Plasmids, fusion proteins, and fungal strains used in this study.**

| Name | Description | Reference |
|---|---|---|
| **Plasmids** | | |
| pPK2-OSCAR-GFP | Construction of gene deletion plasmid | [27] |
| pA-Bar | Construction of gene deletion plasmid | [27] |
| pPK2-Sur-GFP-Ptef | Overexpression of the gene *Coh1* | [2] |
| pPK2-Sur-Ptef-HA | Expression of a protein tagged with HA | [30] |
| pPK2-Bar-Ptef-Myc | Expression of a protein tagged with Myc | This study |
| pPK2-Sur-Ptef-FLAG | Expression of a protein tagged with FLAG | [30] |
| pPK2-Bar-Ptef | For overexpression of *Coh1* in the strain *COH2-N-GFP* | [2] |
| pPK2-NTC-GFP-Ptef | For overexpression of *Coh1* in the strain *COH2-FLAG/COH2-Myc* | [23] |
| **Fusion proteins** | | |
| COH1::HA | COH1 tagged with HA | This study |
| COH2::Myc | COH2 tagged with Myc | This study |
| COH2$^{\Delta DID}$::Myc | Myc-tagged COH2$^{\Delta DID}$ with leucine residues in DID changed into alanine | This study |
| COH2-N::Myc | N-terminus of COH2 tagged with Myc | This study |
| COH2-C::Myc | C-terminus of COH2 tagged with Myc | This study |
| COH2::FLAG | COH2 tagged with FLAG | This study |
| COH2-N::GFP | N-terminus of COH2 tagged with GFP | This study |
| **Fungal strains** | | |
| WT | The wild-type strain of *M. robertsii* ARSEF2575 | |
| *ΔCoh1* | The deletion mutant of the *Coh1* gene | This study |
| *Coh1*$^{OE}$ | The strain overexpressing the *Coh1* gene | This study |
| *C-ΔCoh1* | The complemented strain of the mutant *ΔCoh1* | This study |
| *ΔHat1* | The deletion mutant of the *Hat1* gene | [19] |
| *C-ΔHat1* | The complemented strain of the mutant *ΔHat1* | [19] |
| *ΔHdac1* | The deletion mutant of the *Hda1* gene | This study |
| *Hdac1*$^{OE}$ | The strain overexpressing the *Hdac1* gene | This study |
| *C-ΔHdac1* | The complemented strain of the mutant *ΔHdac1* | This study |
| *ΔCoh2* | The deletion mutant of the *Coh2* gene | This study |
| *C-ΔCoh2* | The complemented strain of the mutant *ΔCoh2* | This study |
| *ΔCoh1::ΔCoh2* | Double deletion mutant of the genes *Coh1* and *Coh2* | This study |
| *ΔCoh2-PCoh2-COH2-FLAG* | Expressing COH2::FLAG driven by *Coh2* promoter in the mutant *ΔCoh2* | This study |
| *WT-Myc* | Expressing the Myc tag in the WT strain | This study |
| *WT-COH2-Myc* | Expressing COH2::Myc in the WT strain | This study |
| *WT-COH2$^{\Delta DID}$-Myc* | Expressing *COH2$^{\Delta DID}$*::Myc in the WT strain | This study |
| *COH1-HA/COH2-Myc* | Expressing COH1::HA and COH2::Myc in the WT strain | This study |
| *COH1-HA/COH2-N-Myc* | Expressing COH1::HA and COH2-N::Myc in the WT strain | This study |
| *COH1-HA/COH2-C-Myc* | Expressing COH1::HA and COH2-C::Myc in the WT strain | This study |
| *WT-FLAG* | Expressing FLAG tag in the WT strain | This study |
| *COH2-FLAG/COH2-Myc* | Expressing COH2::FLAG and COH2::Myc in the WT strain | This study |
| *WT-COH2-N-GFP* | Expressing COH2-N::GFP in the WT strain | This study |
| *Coh1$^{OE}$-COH2-N-GFP* | Expressing COH2-N::GFP in the strain *Coh1$^{OE}$* | This study |
| *Coh1$^{OE}$-COH2-FLAG/Myc* | Expressing Myc tag and COH2::FLAG in the strain *Coh1$^{OE}$* | This study |
| *Coh1$^{OE}$-COH2-FLAG/COH2-Myc* | Expressing COH2::FLAG and COH2::Myc in the strain *Coh1$^{OE}$* | This study |
| *COH1-HA/COH2-N-GFP* | Expressing COH1::HA and COH2-N::GFP in the WT strain | This study |
| *WT-COH2-FLAG* | Expressing COH2::FLAG in the WT strain without the GFP | This study |
| *ΔCoh1-COH2-FLAG* | Expressing COH2::FLAG in the mutant *ΔCoh1* without the GFP | This study |
| *Coh1$^{OE}$-COH2-FLAG* | Expressing COH2::FLAG in the strain *Coh1$^{OE}$* without the GFP | This study |

*(Continued)*

**Table 1.** (Continued)

| Name | Description | Reference |
|---|---|---|
| WT-PHat1-GFP | Expressing GFP driven by *Hat1* promoter in the WT strain | This study |
| WT-PHdac1-GFP | Expressing GFP driven by *Hdac1* promoter in the WT strain | This study |
| WT-PCoh2-GFP | Expressing GFP driven by *Coh2* promoter in the WT strain | This study |
| WT-PDtxS3-GFP | Expressing GFP driven by *DtxS3* promoter in the WT strain | This study |
| WT-PMAA_10199-GFP | Expressing GFP driven by *MAA_10199* promoter in the WT strain | This study |

DID, dimer interface domain; WT, wild-type.

ability of the mutant Δ*Coh1* to penetrate the insect cuticle did not differ from the WT strain (S2I Fig).

We then looked at the involvement of COH1 in fungal interaction with host immune defense. Infection with the WT strain decreased phenoloxidase activity in *G. mellonella* larvae, but no significant difference was found between the mutant Δ*Coh1* and the WT strain (S2J Fig). qRT-PCR analysis showed that infection with the WT strain up-regulated the expression of the antimicrobials gallerimycin and defensin in the insects, but again there was no significant difference between the insects infected by the mutant Δ*Coh1* and the WT strain (S2K Fig).

We also constructed the complemented strain *C-ΔCoh1* of the deletion mutant of *Coh1* by ectopically integrating its genomic clone into the genome of Δ*Coh1* (S1D Fig). Five transformants were randomly selected for the following analysis. In all analyses, no differences were found between the 5 transformants, and data about only 1 transformant is shown. The complemented strain *C-ΔCoh1* was not different from the WT strain in cuticle penetration and $LT_{50}$ values via inoculation by topical application and injection (S3D–S3F Fig). Although the colony growth of the strain *C-ΔCoh1* on PDA plates was not different from the WT strain (S3A Fig), its conidial yield was significantly reduced (S3B Fig). The colony of the strain *C-ΔCoh1* started to autolyze 14 d after inoculation of a conidial suspension on the center of a PDA plate (S3C Fig). In contrast to the WT strain, *Coh1* was expressed in the complemented strain *C-ΔCoh1* during saprophytic growth on the PDA plates (S1E Fig). Therefore, the deletion mutant Δ*Coh1* was not successfully complemented by its genomic clone in the strain *C-ΔCoh1*, suggesting that the native chromosomal position of *Coh1* is essential for its transcriptional regulation. To investigate whether the difference in saprophytic growth on PDA plates between the WT strain and *C-ΔCoh1* resulted from the expression of *Coh1* in *C-ΔCoh1*, we constructed the *Coh1*-overexpressing strain *Coh1^OE* by transforming the coding sequence of COH1, driven by the constitutive promoter *Ptef* from *Aureobasidium pullulans* [18], into the WT strain (S1F Fig). As with *C-ΔCoh1*, *Coh1^OE* showed reduced conidial yield and its colony autolyzed after 14 d of culture on PDA plates (S3B and S3C Fig). The strain *Coh1^OE* was not significantly different from the WT strain and *C-ΔCoh1* in cuticle penetration and $LT_{50}$ values with inoculation via topical application or injection (S3D–S3F Fig).

## Regulation of *Coh1* by a histone deacetylase and a histone acetyltransferase

To identify the mechanisms that regulate *Coh1* expression, we compared the expression level of *Coh1* in the WT strain with that in mutants with regulator genes deleted [2,9,19]. Since it was impossible to collect sufficient hemolymph from the *G. mellonella* larvae for screening a large number of mutants, we searched for the regulators that suppress the expression of *Coh1*

during saprophytic growth in the medium SDY, where *Coh1* was not expressed (Fig 1A). We found that *Coh1* was expressed in 11 epigenetic mutants with deletion of histone acetyltransferases, deacetylases, and methyltransferases, indicating that these epigenetic regulators negatively controlled *Coh1* expression during saprophytic growth (Figs 1C and S4A). Compared with during saprophytic growth, only a histone H3 deacetylase gene (MAA_02098, designated as *Hdac1*) and a histone H3 acetyltransferase gene (MAA_02282, designated as *Hat1*) were significantly down-regulated during surrogate hemocoel colonization (Figs 1D and S4B). No significant differences in the expression levels of *Hat1* and *Hdac1* were found between cuticle penetration and saprophytic growth (Fig 1D). We further compared the expression levels of *Hdac1* and *Hat1* during cuticle penetration with that during colonization of the real hemocoel (i.e., the hemocoel of the insects infected by *M. robertsii*). As we could not obtain enough fungal biomass from the real hemocoel to prepare sufficient RNA for qRT-PCR analysis, we analyzed the expression of *Hdac1* and *Hat1* by tracing the GFP (green fluorescent protein) signal in 2 strains (*WT-PHdac1-GFP* and *WT-PHat1-GFP*), in which the *gfp* gene was driven by *Hdac1* or *Hat1* promoter in the WT strain (S1J Fig). Consistent with the results obtained from the qRT-PCR analysis with the surrogate hemocoel colonization, in both strains the GFP fluorescent intensity in the fungal cells on the cuticle was stronger than in the real hemocoel (Fig 1E).

In a previous study, we constructed a deletion mutant of *Hat1*, Δ*Hat1*, and its complemented strain *C-*Δ*Hat1* [19]. In this study, we generated an *Hdac1* deletion mutant, Δ*Hdac1*, and its complemented strain *C-*Δ*Hdac1* (S1G and S1H Fig). During the surrogate hemocoel colonization, the expression level of *Coh1* was increased 27.6-fold in the mutant Δ*Hat1* and 15.2-fold in the mutant Δ*Hdac1* (Fig 1F). We thus postulated that the reduction in the expression of *Hat1* and *Hdac1* during hemocoel colonization derepressed their negative regulation of *Coh1*, resulting in the expression of *Coh1*. To test this postulation, we first tried to identify the target sites of HAT1 and HDAC1 by immunoblot analysis of the acetylation levels of 7 lysine residues in histone H3. Compared with the WT strain, the acetylation level of histone H3 on lysine 4 (H3K4) was significantly reduced in the mutant Δ*Hat1*, whereas the acetylation level of histone H3 on lysine 56 (H3K56) was increased in the deletion mutants Δ*Hdac1* and Δ*Hat1* (Fig 2A).

We then investigated whether *Hat1* and *Hdac1* regulated the acetylation of histone H3 in the *Coh1* promoter. Using chromatin immunoprecipitation quantitative PCR (ChIP-qPCR) analysis, we found that the acetylation level of histone H3 in the *Coh1* promoter in the WT strain during surrogate hemocoel colonization was 13.8-fold higher than during saprophytic growth. More specifically, the acetylation level of H3K56 in the *Coh1* promoter during surrogate hemocoel colonization was 3.5-fold higher than during saprophytic growth, but no significant difference in the acetylation level of H3K4 was observed between saprophytic growth and surrogate hemocoel colonization. During saprophytic growth, the acetylation level of H3K56 in the *Coh1* promoter in the deletion mutant Δ*Hdac1* was 3.3-fold higher than in the WT strain, but no significant difference was found between the WT strain, the complemented strain *C-*Δ*Hdac1*, and the *Hdac1*-overexpressing strain *Hdac1*$^{OE}$ (Fig 2B). Overexpression of *Hdac1*, driven by the constitutive promoter *Ptef* from *A. pullulans* [18], was confirmed by qRT-PCR (S1I Fig). During surrogate hemocoel colonization, H3K56 in the promoter of *Coh1* in the WT strain was 5-fold more acetylated than in the strain *Hdac1*$^{OE}$, but no difference was found among the WT strain, the deletion mutant Δ*Hdac1*, and the strain *C-*Δ*Hdac1* (Fig 2B).

As described above, HAT1 is a histone acetyltransferase. Unexpectedly, the acetylation levels of histone H3, H3K4, and H3K56 in the *Coh1* promoter in the WT strain were all lower than in the mutant Δ*Hat1*; however, no significant differences were observed between the WT strain and the strain *C-*Δ*Hat1* (Fig 2C). It is likely that HAT1 does not directly acetylate histone

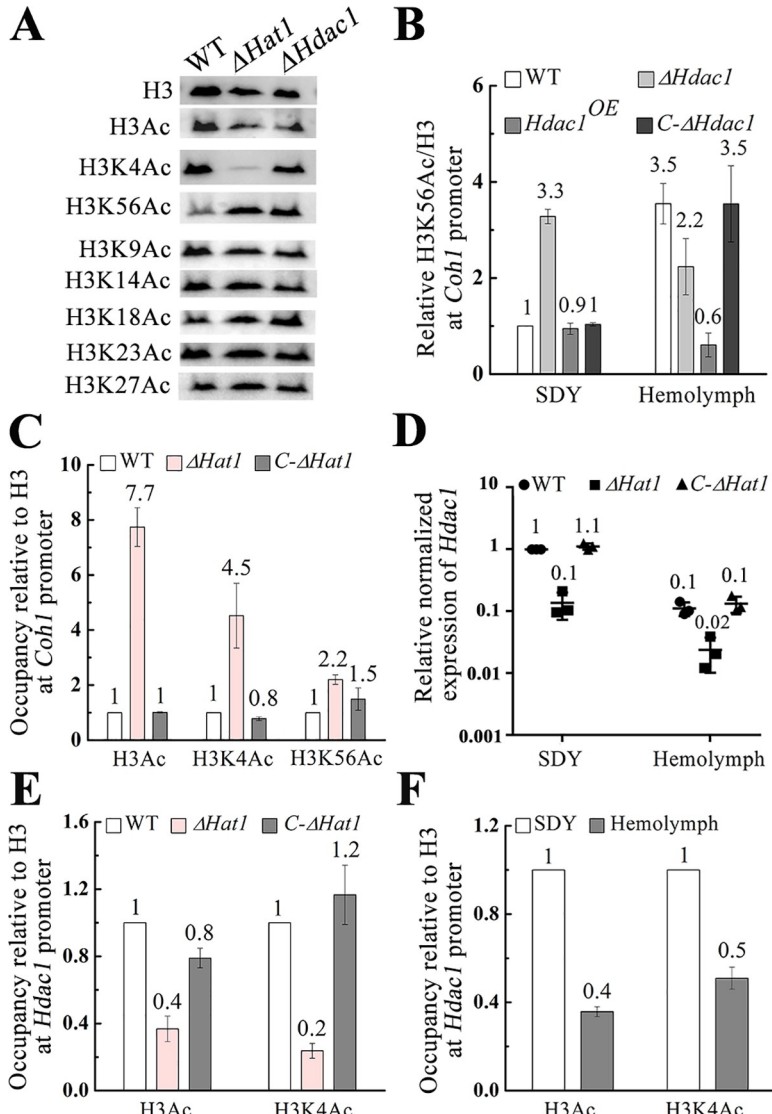

**Fig 2. HDAC1 and HAT1 control *Coh1* expression by regulating histone H3 acetylation in its promoter region.** (A) Immunoblot analysis of acetylation levels of 7 lysine residues on histone H3 protein in the wild-type (WT) strain and the mutants Δ*Hdac1* and Δ*Hat1*. All Western blot images shown in this study are representatives of at least 3 independent experiments. (B) The acetylation level of H3K56 in the *Coh1* promoter in the mutant Δ*Hdac1*, its complemented strain *C-*Δ*Hdac1*, the *Hdac1*-overexpressing strain *Hdac1*^OE, and the WT strain during saprophytic growth in Sabouraud dextrose broth supplemented with 1% yeast extract (SDY) and surrogate hemocoel colonization (Hemolymph). (C) The acetylation levels of histone H3, H3K56, and H3K4 in the *Coh1* promoter in the mutant Δ*Hat1* and its complemented strain *C-*Δ*Hat1* relative to the WT strain during saprophytic growth. (D) qRT-PCR analysis of *Hdac1* expression in the WT strain, the mutant Δ*Hat1*, and its complemented strain *C-*Δ*Hat1* during saprophytic growth and surrogate hemocoel colonization. (E) The acetylation levels of histone H3 and H3K4 in the *Hdac1* promoter in the mutant Δ*Hat1* and its complemented strain *C-*Δ*Hat1* relative to the WT strain during saprophytic growth. (F) The acetylation levels of histone H3 and H3K4 in the *Hdac1* promoter in the WT strain during surrogate hemocoel colonization relative to saprophytic growth. For chromatin immunoprecipitation quantitative PCR (ChIP-qPCR) analyses in this figure, the values represent the fold-change of the acetylation level of histone H3, H3K4, or H3K56 compared with the level in its respective control, which is set to 1. All ChIP-qPCR experiments were repeated at least 3 times. The data underlying all the graphs shown in this figure can be found in S1 Data.

H3 in the promoter of *Coh1* but instead could regulate other components, which in turn alter the histone H3 acetylation level in the *Coh1* promoter. Since the target of HDAC1 is H3K56, and HAT1 negatively regulated the global acetylation level of H3K56 (Fig 2A), we investigated

whether HAT1 regulated *Hdac1* expression by controlling the acetylation of histone H3 in its promoter. qRT-PCR analysis showed that the expression level of *Hdac1* in the WT strain was significantly higher than in the mutant Δ*Hat1* during saprophytic growth and surrogate hemocoel colonization (Fig 2D), suggesting that HAT1 positively regulated *Hdac1* transcription. ChIP-qPCR analysis further showed that the acetylation levels of histone H3 and H3K4 in the *Hdac1* promoter in the mutant Δ*Hat1* were both lower than in the WT strain (Fig 2E). In the WT strain, the acetylation levels of histone H3 and H3K4 in the promoter of *Hdac1* during saprophytic growth were significantly higher than during surrogate hemocoel colonization (Fig 2F). To further confirm that HAT1 regulated the expression of *Hdac1*, which in turn controlled *Coh1* expression, we constructed the strain Δ*Hat1-Hdac1*$^{OE}$ with *Hdac1* overexpressed in the mutant Δ*Hat1* (S1I Fig). qRT-PCR analysis showed that the expression level of *Coh1* in Δ*Hat1* during surrogate hemocoel colonization was 3.3-fold higher than that of Δ*Hat1-Hdac1*$^{OE}$ (S4C Fig).

## COH1 physically interacts with the transcription factor COH2

As a putative regulatory protein, COH1 could interact with other proteins to control hemocoel colonization. Using a pull-down assay with the strain *WT-COH1-HA* expressing the fusion protein COH1::HA (a protein with COH1 tagged with HA [hemagglutinin]) in the WT strain (S5B Fig), we failed to identify the proteins that interact with COH1. Unless otherwise indicated, all genes encoding fusion proteins were driven by the constitutive promoter *Ptef* from *A. pullulans* in this study. Previous studies showed that alteration of expression of a gene can change the expression pattern of its interacting components [20–23], so we used RNA-Seq to profile the differentially expressed genes (DEGs) between the *Coh1*-overexpressing strain *Coh1*$^{OE}$ and the WT strain to identify candidates interacting with COH1. There were 1,893 DEGs, with 1,227 genes up-regulated and 666 genes down-regulated in the *Coh1*$^{OE}$ strain. As a regulatory protein, it is more likely that COH1 interacts with other regulators to control gene expression. Compared with the WT strain, 5 transcription factors were up-regulated in the strain *Coh1*$^{OE}$. Yeast 2-hybrid assays showed that COH1 interacted with 1 transcription factor (MAA_07838) (Fig 3A), which is designated as COH2, as it also regulates colonization of the hemocoel (see below). *Coh2* has an 849-bp ORF that encodes a protein containing 282 amino acid residues with a deduced molecular weight of 30.9 kDa. COH2 is a bZIP transcription factor with a DNA binding domain (COH2-DBD; Ser-30 to Thr-50), NLS (Ser-16 to Asp-39), and dimer interface domain (DID) (His-53 to Leu-88).

We further conducted a coimmunoprecipitation (Co-IP) assay using a strain (*COH1-HA/COH2-Myc*) constitutively expressing COH1::HA and COH2::Myc (a protein with COH2 tagged with Myc) and confirmed that COH1 physically interacted with COH2 in vivo (Fig 3B). In the Co-IP assay, the Western blot analysis with an anti-Myc antibody always showed that the mass of the detected protein was approximately 2-fold greater than the predicted molecular weight (44.6 kDa) of the protein COH2::Myc (Fig 3B), suggesting that COH2 formed a homodimer. To confirm this, a further Co-IP assay using a strain (*COH2-FLAG/COH2-Myc*) constitutively expressing COH2::FLAG (a protein with FLAG fused to COH2) and COH2::Myc showed that these 2 fusion proteins formed a dimer in vivo (S6A Fig). The DID in COH2 contains 7 leucine residues that could be responsible for the dimerization of this transcription factor. To test this, we constructed a strain (*COH2*$^{ΔDID}$-*Myc*) expressing the protein COH2$^{ΔDID}$::Myc with the leucine residues in the DID substituted to alanine in the fusion protein COH2::Myc. Western blot analysis showed that the fusion protein COH2$^{ΔDID}$::Myc did not form dimers (S6B Fig).

We then assayed the region of COH2 that physically contacted COH1 using Co-IP assays. COH2 was divided into the N-terminus section (Met-1 to Thr-100) and the C-terminus section

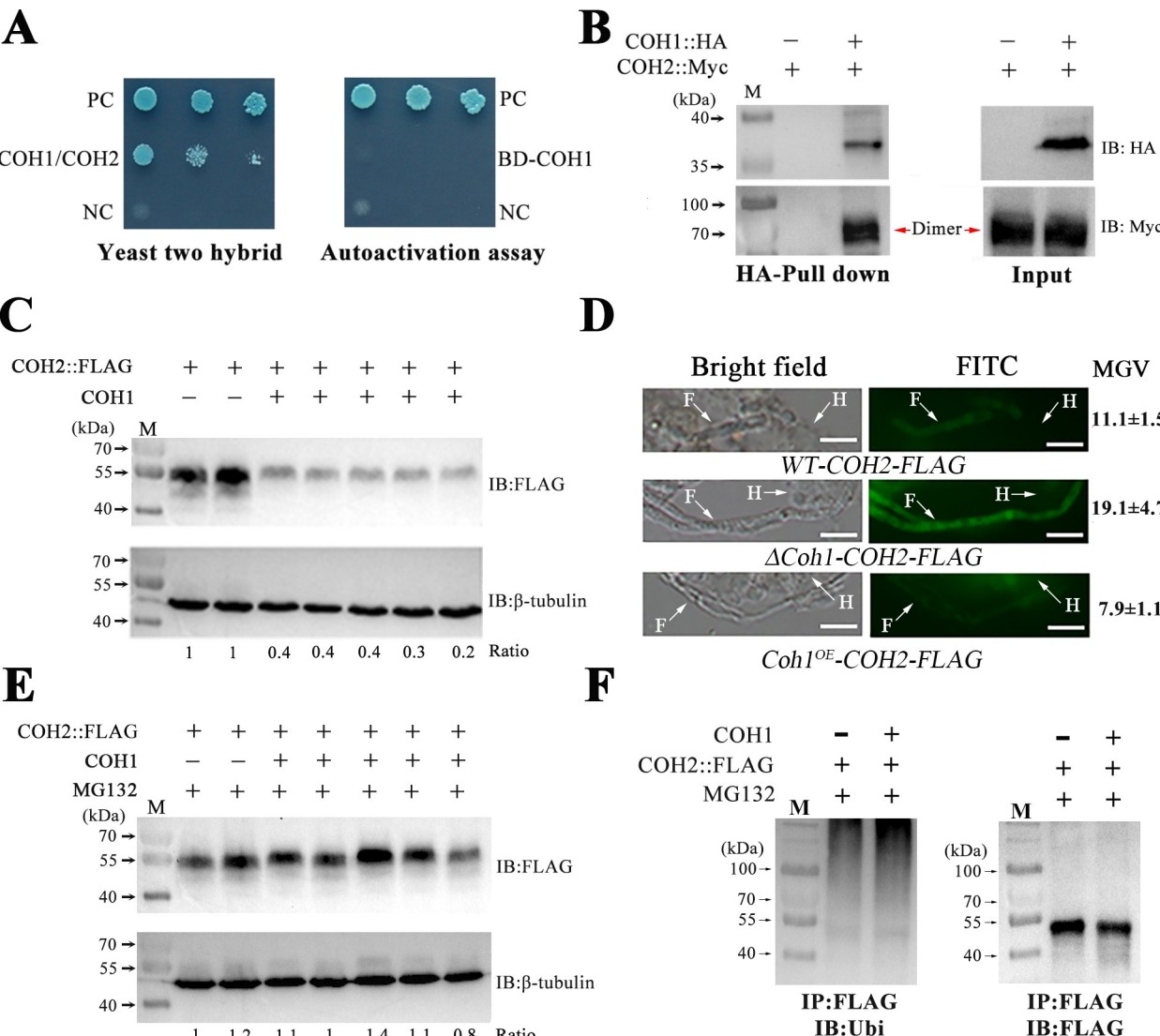

**Fig 3. COH1 physically interacts with the transcription factor COH2 to reduce COH2 stability.** (A) Yeast 2-hybrid analysis confirms the physical interaction of COH1 with COH2. Left panel: colonies were grown in SD/−Ade/−His/−Leu/−Trp + X-α-gal + AbA. Right panel: COH1 lacks autoactivation activity. The Y2HGold cells with pGBKT7-COH1 cannot grow in SD/−Ade/−His/−Trp + X-α-gal. BD, binding domain; NC, negative control (yeast cells containing the plasmid pGADT7-T and pGBKT7-Lam); PC, positive control (yeast cells containing the plasmid pGADT7-T and pGBKT7-53). (B) Coimmunoprecipitation confirmation of the physical interaction of COH1 with COH2. The fusion proteins COH1::HA and COH2::Myc (molecular weight = 44.6 kDa) were simultaneously expressed in the strain *COH1-HA/COH2-Myc*. The control was the strain *WT-COH2-Myc* expressing the protein COH2::Myc. Immunoprecipitation was conducted with anti-HA antibody. Proteins were detected by immunoblot (IB) analysis with anti-HA or anti-Myc antibodies. The dimer COH2::Myc is indicated by red arrows. M, protein ladder. (C) Differential accumulation of the COH2::FLAG protein in 2 isolates of the strain *WT-COH2-FLAG* and 5 isolates of the strain *Coh1^{OE}-COH2-FLAG*. Equal loading of proteins was confirmed by the β-tubulin protein that was detected by the anti-β-tubulin antibody. Numbers indicate band intensity for COH2::FLAG relative to β-tubulin. The values of the #1 isolate of the strain *WT-COH2-FLAG* were set to 1. (D) Histoimmunochemical staining of the COH2::FLAG protein in fungal cells in the real hemocoel of *G. mellonella* larvae. Top panel: the strain *WT-COH2-FLAG*; middle panel: Δ*Coh1-COH2-FLAG*; bottom panel: *Coh1^{OE}-COH2-FLAG*. Scale bar represents 10 μm. Images are representative of 3 independent experiments. F, fungal cells; FITC, fluorescein isothiocyanate; H, hemocyte; MGV, mean gray value. (E) Confirmation of degradation of the COH2::FLAG protein by the proteasome pathway. The mycelium grown in Sabouraud dextrose broth supplemented with 1% yeast extract (SDY) medium was treated with the 26S proteasome inhibitor MG132. (F) The ubiquitination level of the COH2::FLAG protein increased due to its interaction with COH1. The COH2::FLAG protein was pulled down from the MG132-treated mycelium with an anti-FLAG antibody and immunoblotted with an anti-ubiquitin (Ubi) antibody (left) and the anti-FLAG antibody (right). IP, immunoprecipitation.

(Ser-101 to Arg-282). The N-terminus section contained the predicted DNA binding domain COH2-DBD, the DID, and the NLS. For Co-IP assays, we constructed 2 strains: *COH1-HA/ COH2-N-Myc*, expressing HA-tagged COH1 and the fusion protein with the N-terminus of COH2 (COH2-N) fused with Myc, and *COH1-HA/COH2-C-Myc*, expressing the HA-tagged COH1 and the COH2 C-terminus (COH2-C) fused with Myc. Co-IP assays showed that COH1 can physically interact with the N-terminus of COH2, but did not with the C-terminus (S6C and S6D Fig). As with the protein COH2::Myc, COH2-N::Myc also formed dimers (S6C Fig).

The protein COH2::FLAG was used as a representative of the COH2 fusion proteins used in this study (see below) to assay whether the COH2 protein fused with a tag retained its WT activity. To this end, we constructed a COH2::FLAG expression cassette with its coding sequence driven by the *Coh2* promoter, which was then transformed into the mutant Δ*Coh2* (see below) to produce the strain Δ*Coh2-PCoh2-COH2-FLAG*. No differences in saprophytic growth and pathogenicity were found between the WT strain, Δ*Coh2-PCoh2-COH2-FLAG*, and the complemented strain *C-ΔCoh2*, indicating that the COH2::FLAG protein functioned as COH2 (S7A and S7B Fig). However, this method could not be used to investigate whether the COH1 fusion proteins used in this study retained their WT activity because the native chromosomal position of *Coh1* is essential for its promoter activity (S1D and S1E Fig). However, the *Coh1*-overexpressing strain *Coh1*$^{OE}$ was not different in colony growth and pathogenicity from the strain constitutively expressing the fusion protein COH1::Myc (S3C Fig), indirectly showing that COH1 and COH1::Myc had the same functions.

## The COH1 and COH2 interaction reduces COH2 stability

The impacts of the physical interaction between COH1 and COH2 on the activity of the transcription factor COH2 were then investigated. We first assayed the impact of the interaction on COH2 stability. To this end, the protein level of the fusion protein COH2::FLAG in the strain *WT-COH2-FLAG* (the protein COH2::FLAG expressed in the WT strain) was compared with the strain *Coh1*$^{OE}$*-COH2-FLAG* (COH2::FLAG expressed in the *Coh1*-overexpressing strain *Coh1*$^{OE}$). Five isolates of the strain *Coh1*$^{OE}$*-COH2-FLAG* and 2 isolates of the strain *WT-COH2-FLAG* were selected, as they had the same transcription level of the fusion gene encoding the protein COH2::FLAG (S8A Fig). In the SDY medium, where COH1 was expressed in the strain *Coh1*$^{OE}$*-COH2-FLAG* but not in *WT-COH2-FLAG*, 2 isolates of the strain *WT-COH2-FLAG* both accumulated much higher levels of the protein COH2::FLAG than all 5 isolates of the strain *Coh1*$^{OE}$*-COH2-FLAG* (Fig 3C). We further assayed whether COH1 reduced the amount of the COH2::FLAG protein in the real hemocoel of insects infected by *M. robertsii*. With histoimmunochemical staining of the fungal cells collected from the real hemocoel with anti-FLAG antibody, the COH2::FLAG protein was found to be more abundant in the strain Δ*Coh1-COH2-FLAG* than the strain *WT-COH2-FLAG* (*Coh1* expressed in the real hemocoel), which in turn had more COH2::FLAG protein than the *Coh1*-overexpressing strain *Coh1*$^{OE}$*-COH2-FLAG* (Fig 3D). To investigate whether reduction of the COH2::FLAG protein caused by the expression of COH1 was due to degradation by the proteasome, the strains *WT-COH2-FLAG* and *Coh1*$^{OE}$*-COH2-FLAG* were grown in the SDY medium supplemented with MG132, a specific inhibitor of the 26S proteasome. As shown in Fig 3E, MG132 treatment increased the level of the protein COH2::FLAG in the strain *Coh1*$^{OE}$*-COH2-FLAG*, but not in the strain *WT-COH2-FLAG*, indicating that the interaction between COH1 and COH2 induces the degradation of the COH2::FLAG protein by the proteasome. We further found that the ubiquitination level of the COH2::FLAG protein in the strain *Coh1*$^{OE}$*-COH2-FLAG* was higher than in *WT-COH2-FLAG*, indicating that the ubiquitin–proteasome pathway was responsible for the COH2 degradation (Fig 3F).

We then tested the impact of COH1 on COH2 dimerization. To do this, we constructed a strain (*Coh1^OE^-COH2-FLAG/COH2-Myc)* with the fusion proteins COH2::FLAG and COH2:: Myc expressed in the strain *Coh1^OE^* (S5 Fig). As in the strain *COH2-FLAG/COH2-Myc*, with the fusion proteins COH2::FLAG and COH2::Myc expressed in the WT strain (S5 Fig), COH2 dimerized in the strain *Coh1^OE^-COH2-FLAG/COH2-Myc* (S8B Fig), showing that COH1 did not impact COH2 dimerization.

We also assayed the impact of COH1 on the entry of COH2 into the nucleus. To this end, we tried to construct a strain expressing the fusion protein COH2::GFP in the strain *Coh1^OE^*, but this attempt was not successful because the protein COH2::GFP was always separated into the 2 proteins: COH2 and GFP. However, we successfully expressed a fusion protein COH2-N::GFP (GFP fused to the COH2 N-terminus containing the NLS, COH2-DBD, and DID) in the WT strain and the strain *Coh1^OE^* to produce *WT-COH2-N-GFP* and *Coh1^OE^-COH2-N-GFP* (S5C Fig), respectively. In these 2 strains during saprophytic growth in SDY medium and during real hemocoel colonization, the GFP fluorescence intensity was strongest in the nucleus (S8C Fig). However, using the strain *COH1-HA/COH2-N-GFP* that simultaneously expressed COH1::HA and COH2-N::GFP, a Co-IP analysis showed that these 2 fusion proteins could not physically contact each other (S8D Fig). In addition, the protein COH2-N:: GFP did not undergo dimerization (S5C Fig). Therefore, COH2-N::GFP is not suitable for assaying the impact of COH1 on the entry of COH2 into the nucleus.

## COH2 is involved in pathogenicity

To investigate the biological functions of *Coh2*, we first assayed its expression pattern. qRT-PCR analysis showed that *Coh2* was constitutively expressed during saprophytic growth, cuticle penetration, and surrogate hemocoel colonization (S9A Fig). For the strain *WT-PCoh2-GFP*, with *gfp* driven by *Coh2* promoter in the WT strain, the GFP fluorescent signal in the fungal cells on the insect cuticle was as strong as that in the real hemocoel of infected insects (S9B Fig), showing that *Coh2* was constitutively expressed during cuticle penetration and real hemocoel colonization.

We then constructed a *Coh2* deletion mutant (Δ*Coh2*) and its complemented strain *C-Δ*Coh2* (S1G and S1H Fig). On PDA plates, no significant differences in colony growth, colony phenotype, or conidial yield were seen between Δ*Coh2*, *C-Δ*Coh2*, and the WT strain (S9C and S9D Fig). With topical application of conidia on the insect cuticle, the $LT_{50}$ value of Δ*Coh2* (17.4 ± 0.59 d) was 2-fold higher than that of the WT strain (8.7 ± 0.05 d) *(P < 0.05)*, but no significant difference *(P > 0.05)* was found between the WT strain and the complemented strain *C-Δ*Coh2* (8.6 ± 0.07 d). With direct injection, the $LT_{50}$ value of Δ*Coh2* (4.5 ± 0.24 d) was slightly higher than that of the WT strain (3.6 ± 0.09 d) *(P < 0.05)*, and again the strain *C-Δ*Coh2* (3.7 ± 0.13 d) was not significantly different *(P > 0.05)* from the WT strain. On the hydrophobic surfaces of plastic petri dishes, appressorial formation was delayed in the mutant Δ*Coh2*, whereas no significant difference was observed between the WT strain and *C-Δ*Coh2* (S9E Fig). Compared with the WT strain, the ability of the mutant Δ*Coh2* to penetrate the cuticle was decreased, and again no difference was found between the WT strain and *C-Δ*Coh2* (S9F Fig).

To investigate how *Coh1* interacted with *Coh2* to regulate pathogenicity, we constructed the double gene deletion mutant Δ*Coh1*::Δ*Coh2* (S1G Fig). *WT-COH2-FLAG* and Δ*Coh1-COH2-FLAG* were strains that overexpressed *Coh2* in the WT strain and the mutant Δ*Coh1*, respectively. With topical application, the $LT_{50}$ value of the mutant Δ*Coh1*::Δ*Coh2* was not significantly different from those of Δ*Coh1* and Δ*Coh2*, but was significantly higher than those of the strains *WT-COH2-FLAG* and Δ*Coh1-COH2-FLAG*. Compared to the WT

strain, the virulence of the strains *WT-COH2-FLAG* and *ΔCoh1-COH2-FLAG* was also significantly reduced (S9G Fig). With direct injection, the $LT_{50}$ value of the strain *ΔCoh1::ΔCoh2* was not significantly different from those of *ΔCoh1*, *ΔCoh2*, *WT-COH2-FLAG*, and *ΔCoh1-COH2-FLAG* (S9H Fig). With both inoculation methods, no significant difference in fungal burden was found between the insects infected by *ΔCoh1*, *ΔCoh2*, *ΔCoh1::ΔCoh2*, *WT-COH2-FLAG*, and *ΔCoh1-COH2-FLAG*, which had significantly lower fungal burden than insects infected with the WT strain (S2F and S2G Fig). Compared to insects infected by the WT strain, the phenoloxidase activity in insects infected by the mutant *ΔCoh2* was increased 2.3-fold ($P < 0.05$). No significant difference in phenoloxidase activity was found between the insects infected by the WT strain, *ΔCoh1*, *C-ΔCoh2*, *WT-COH2-FLAG*, and *ΔCoh1-COH2-FLAG* (S2J Fig). No significant differences in the expression levels of the antimicrobials *gallerimycin* and *defensin* were found between the insects infected by the WT strain and the mutants *ΔCoh1*, *ΔCoh2*, *WT-COH2-FLAG*, and *ΔCoh1-COH2-FLAG* (S2K Fig).

## Determination of the DNA motif bound by COH2

To investigate how the transcription factor COH2 regulates pathogenicity genes with the regulatory protein COH1, we first used ChIP-Seq analysis with the strain *WT-COH2-FLAG* to identify the genes directly regulated by COH2. In the ChIP-Seq analysis, the peak caller MACS (Model-based Analysis of ChIP-Seq) identified a total of 562 peaks that were bound by COH2 (S10A Fig), 66.4% of which contained a consensus 7-nucleotide motif (TGA[C/G]T[C/A][G/A]) with multiple possible nucleotides at position 4, 6, and 7 (Figs 4A and S10B). This motif is designated as *COH2-BM* (COH2 binding motif). Consistent with COH2 forming a dimer, the motif *COH2-BM* contains a palindrome sequence: TGA and TCA spaced by 1 nucleotide at the fourth position. Electrophoretic mobility shift assay (EMSA) was then used to confirm the binding of COH2 to the motif *COH2-BM* (TGAGTCT) in the promoter of the gene MAA_04430, a representative of genes with the motif *COH2-BM* in their promoters determined by the ChIP-Seq analysis. We failed to express the whole protein of COH2 in *Escherichia coli*, but the portion (Met-1 to Thr-100) containing the predicted DNA binding domain, designated as COH2-DBD, was successfully expressed and the recombinant protein was then purified for EMSA (S10C and S10D Fig). The gel shift assay showed the DNA probe containing the motif *COH2-BM* (biotin-labeled) bound to the recombinant COH2-DBD protein (Fig 4B). The specific competitor (unlabeled DNA probe) almost completely abolished the DNA band shift (Fig 4B). To assay the importance of each position in the motif *COH2-BM* for binding to COH2-DBD, unlabeled *COH2-BM* mutants were constructed by replacing the consensus base with each of the other 3, and these mutants were subsequently used as competitors for the binding of the biotin-labeled DNA probe to the recombinant protein COH2-DBD. Consistent with the data obtained from the ChIP-Seq analysis, only the probe with G to A mutation at position 7 almost completely abolished the band shift of the labeled DNA probe, and the probe with C to A mutation at position 6 and all 3 mutated probes at the space nucleotide (at position 4) also significantly altered the band shift (S10E and S10F Fig). Mutations at other positions also impacted the band shift, but to a lesser extent than the mutated probes described above (S10F Fig).

To validate the ChIP-Seq and EMSA results, we performed ChIP-qPCR with primers designed to cover the motif *COH2-BM* in the promoter of the gene MAA_04430. After immunoprecipitation, qPCR analysis showed that the copy number of the DNA fragment containing the motif *COH2-BM* from the strain *WT-COH2-FLAG* was 29.4-fold higher than that from the strain *WT-FLAG*, which expressed the FLAG tag only (Fig 4C).

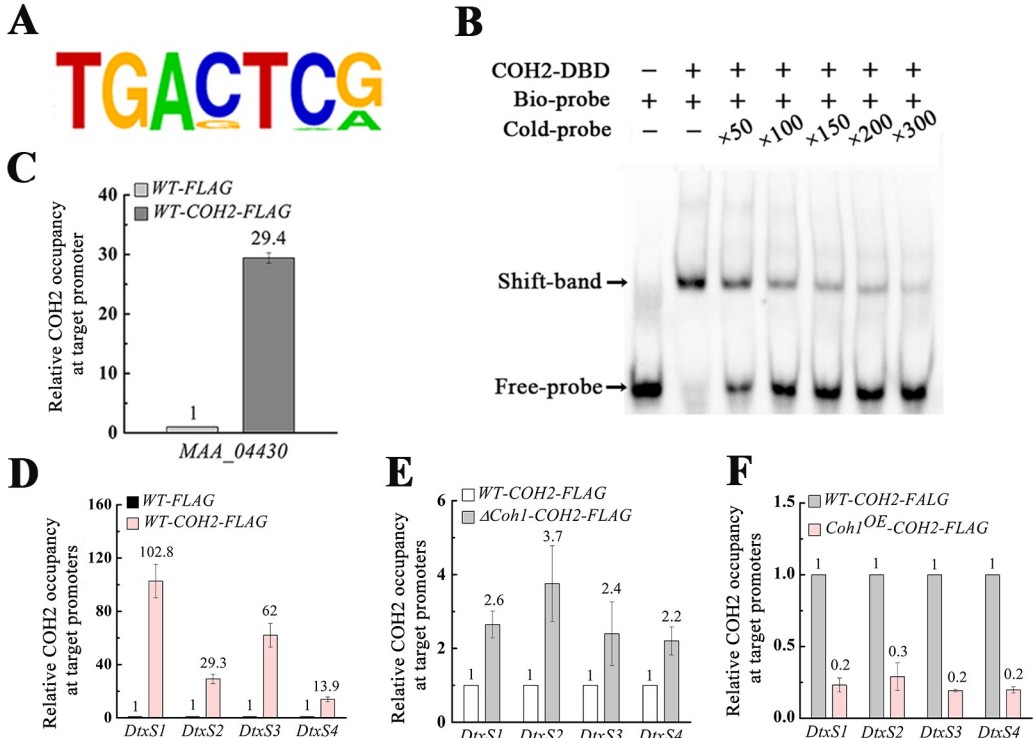

**Fig 4. Identification of hemocoel-colonizing genes regulated by COH1 and COH2.** (A) Chromatin immunoprecipitation sequencing (ChIP-Seq) analysis identified the DNA motif *COH2-BM* that was bound by the transcription factor COH2. (B) Electrophoretic mobility shift assay (EMSA) confirms the in vitro binding of the biotin-labeled motif *COH2-BM* (Bio-probe) to the recombinant protein COH2-DBD (COH2 DNA binding domain). The binding activity was demonstrated by the shift of the labeled DNA band prior to the addition of the specific competitor (Cold-probe: the unlabeled motif *COH2-BM*) in 50-, 100-, 150-, 200-, or 300-fold excess. The tested DNA motif *COH2-BM* is from the promoter of the gene MAA_04430, which was shown to have the motif *COH2-BM* by the ChIP-Seq analysis. (C) Chromatin immunoprecipitation quantitative PCR (ChIP-qPCR) analysis confirms that COH2 in vivo binds to the motif *COH2-BM* in the MAA_04430 promoter in the strain *WT-COH2-FLAG*, expressing the fusion protein COH2::FLAG. *WT-FLAG*: a strain expressing the tag FLAG only. (D) ChIP-qPCR confirmation of the in vivo binding of COH2 to the promoters of the 4 destruxin biosynthesis genes (*DtxS1*, *DtxS2*, *DtxS3*, and *DtxS4*). (E) ChIP-qPCR analysis shows that during surrogate hemocoel colonization, deleting the *Coh1* gene increased the binding of COH2 to the promoters of the 4 destruxin biosynthesis genes. *ΔCoh1-COH2-FLAG*: a strain expressing the protein COH2::FLAG in the mutant *ΔCoh1*. (F) ChIP-qPCR analysis confirms that during saprophytic growth, COH1 reduced the in vivo binding of COH2 to the promoters of the 4 destruxin biosynthesis genes. *Coh1^{OE}-COH2-FLAG*: a strain expressing the protein COH2::FLAG and overexpressing the *Coh1* gene. The data underlying all the graphs shown in this figure can be found in S1 Data.

## The COH1 and COH2 interaction derepresses COH2-mediated repression of hemocoel colonization genes

As COH1 was only expressed during hemocoel colonization, we thus investigated how COH1 and COH2 interacted to regulate this infection stage. To this end, RNA-Seq analysis was first used to compare the transcriptomes of the WT strain and the mutants *ΔCoh1* and *ΔCoh2* during surrogate hemocoel colonization. Compared with the WT strain, 68 genes were up-regulated and 207 genes down-regulated in the mutant *ΔCoh2* (S11A Fig). Compared with the WT strain, 52 genes were down-regulated and 97 genes up-regulated in the mutant *ΔCoh1* (S11A Fig). RNA-Seq analysis showed that neither *Coh1* nor *Coh2* regulated *Hat1*expression, and subsequent qRT-PCR analysis further confirmed that there was no significant difference in the expression level of *Hat1* between the WT strain and the mutants *ΔCoh1* and *ΔCoh2*, and between the *Coh1*-overexpressing strain *Coh1^{OE}* and the *Coh2*-overexpressing strain

*WT-COH2-FLAG* (S11C Fig). Remarkably, among the 15 genes that were down-regulated in Δ*Coh1* but up-regulated in Δ*Coh2* (S11B Fig), 10 were involved in hemocoel colonization, including 4 genes in the destruxin biosynthesis pathway, a siderophore iron transporter (MAA_10457), and a laccase (MAA_00990). To investigate whether these 15 genes are directly regulated by COH2, we searched for the motif *COH2-BM* in their promoters, which were determined as the approximately 2-kb DNA fragments upstream of the ORF start sites or as the regions, if shorter than 2 kb, between the ORFs of the 15 genes and the ORFs of their respective adjacent genes. The motif *COH2-BM* was found in the promoters of all the 15 genes.

Using the 4 destruxin biosynthesis genes (*DtxS1*, *DtxS2*, *DtxS3*, and *DtxS4*) as representatives, we investigated the mechanisms for COH1 and COH2 to regulate the genes involved in hemocoel colonization. qRT-PCR further confirmed that during surrogate hemocoel colonization, compared with the WT strain, these 4 destruxin biosynthesis genes were down-regulated in the mutant Δ*Coh1* and up-regulated in the mutant Δ*Coh2* (S11D Fig). Using the gene *DtxS3* as a representative, we assayed the impact of *Coh1* and *Coh2* on the expression of the destruxin biosynthesis genes in the real hemocoel of insects infected by *M. robertsii*. To this end, an expression cassette with *gfp* driven by the *DtxS3* promoter *PDtxS3* was transformed into the WT strain and the mutants Δ*Coh1* and Δ*Coh2* to produce the strains *WT-PDtxS3-GFP*, Δ*Coh1-PDtxS3-GFP*, and Δ*Coh2-PDtxS3-GFP* (S1J Fig). GFP fluorescent intensity in Δ*Coh2-PDtxS3-GFP* cells collected from the real hemocoel appeared to be stronger than in *WT-PDtxS3-GFP*, which was in turn stronger than in Δ*Coh1-PDtxS3-GFP* (S11E Fig).

ChIP-qPCR analysis showed that COH2 occupancy at *DtxS1*, *DtxS2*, *DtxS3*, and *DtxS4* promoters in the strain *WT-COH2-FLAG* was 102.8-, 29.3-, 62.0-, and 13.9-fold higher, respectively, than in the strain *WT-FLAG* (Fig 4D). Based on the qRT-PCR and ChIP-qPCR data described above, we postulated that COH2 is a repressor of the destruxin biosynthesis genes and that this repression is mitigated by COH1 when the fungus enters the insect hemocoel. To test this, we assayed the impact of the presence of COH1 on the binding of COH2 to the promoters of the 4 genes. To this end, we constructed a strain (Δ*Coh1-COH2-FLAG*) with the fusion protein COH2::FLAG expressed in the mutant Δ*Coh1* (S5A Fig). During surrogate hemocoel colonization, the enrichments of the protein COH2::FLAG in the promoters of *Dtxs1*, *Dtxs2*, *Dtxs3*, and *Dtxs4* in the strain Δ*Coh1-COH2-FLAG* were all higher than in *WT-COH2-FLAG* (Fig 4E). Conversely, compared with the strain *WT-COH2-FLAG*, the enrichment of COH2::FLAG at the promoter of *Dtxs1*, *Dtxs2*, *Dtxs3*, and *Dtxs4* was reduced 5.3-, 3.4-, 5.3-, and 5.3-fold, respectively, in the strain *Coh1^{OE}-COH2-FLAG* (Fig 4F).

## COH2 induces the expression of cuticle-degrading genes

As described above, in addition to regulating hemocoel colonization along with the regulator COH1, COH2 also controlled cuticle penetration. To investigate how COH2 regulates cuticle penetration, we used RNA-Seq to compare the transcriptomes of the mutant Δ*Coh2* and the WT strain when they were grown on locust hindwings. Compared with the WT strain, 549 genes were down-regulated and 236 genes up-regulated in Δ*Coh2* (S11A Fig). The COH2 binding motif *COH2-BM* was found in 368 genes out of the 549 down-regulated genes, and 171 out of the 236 up-regulated genes. In this study, the *G. mellonella* larva is used as the host for pathogenicity analysis. Therefore, by qRT-PCR analysis of 5 representative genes, we validated the results obtained with locust hindwings on the cuticle of *G. mellonella* larva, and no difference in their expression was found between the locust hindwings and the *G. mellonella* cuticle (S11G Fig).

Remarkably, 44 cuticle-degrading genes were down-regulated in the deletion mutant Δ*Coh2*, including 30 proteases, 3 chitinases, 2 lipases, and 9 cytochrome P450 enzymes

(S11F Fig). Among the 30 protease genes, 27 had the motif *COH2-BM* in their promoters. The *COH2-BM* motif was also found in the promoters of the 3 chitinase genes, 1 lipase gene, and 7 cytochrome P450 genes. Two protease genes (MAA_10199 and MAA_10350) and 1 chitinase gene (MAA_10456) were used as representatives to confirm that COH2 directly regulated the cuticle-degrading genes described above. During penetration of the *G. mellonella* cuticle, qRT-PCR confirmed that these 3 genes were down-regulated in the mutant Δ*Coh2* compared with the WT strain (Fig 5A). Since it was not possible to obtain enough biomass to get sufficient DNA for ChIP-qPCR analysis during cuticle penetration, i.e., growing *M. robertsii* on the insect cuticle, an approximation of cuticle penetration was prepared by growing the fungus in a cuticle medium with the cuticle of *G. mellonella* larvae as the sole carbon and nitrogen source (called cuticle medium hereafter). When fungus was grown in the cuticle medium, ChIP-qPCR analysis showed that in the strain *WT-COH2-FLAG*, the COH2::FLAG protein bound to the promoters of these 3 genes (Fig 5B).

We further investigated the impact of the cuticle-degrading genes regulated by COH2 on the ability of the fungus to degrade the insect cuticle. To this end, we quantified cuticle degradation products following 12 h of fungal growth in the cuticle medium. Although the growth rates of the WT strain and the mutant Δ*Coh2* in the cuticle medium were similar, the mutant secreted significantly less total extracellular protease than the WT strain ($P < 0.05$) (S11H Fig), and it thus released significantly fewer amino acids (S11I Fig) and peptides (S11J Fig) from the cuticle ($P < 0.05$). Likewise, the WT strain produced significantly ($P < 0.05$) more chitinase than Δ*Coh2* (S11K Fig).

## COH1 represses expression of cuticle-degrading genes during hemocoel colonization

The expression of cuticle-degrading genes is inhibited once the fungus enters the insect hemocoel, otherwise they cause hyperactive immunity of the insects, which reduces fungal reproducibility in the insect cadavers [9,24]. We thus investigated whether COH2-mediated induction of the cuticle-degrading genes was inactivated by COH1 when the fungus entered the hemocoel. The 2 proteases MAA_10199 and MAA_10350 and the chitinase MAA_10456 were again used as representatives for cuticle-degrading genes regulated by COH2 during cuticle penetration. Compared with cuticle penetration, the expression levels of the 3 cuticle-degrading genes were all significantly reduced during surrogate hemocoel colonization (Fig 5C). Consistent with the expression patterns of these the 3 genes, ChIP-qPCR analysis showed the that in vivo binding of COH2 to their promoter regions during growth in the cuticle medium was significantly greater than during surrogate hemocoel colonization (Fig 5D). During surrogate hemocoel colonization, the expression levels of these 3 cuticle-degrading genes in the mutant Δ*Coh1* was higher than in the WT strain (Fig 5E), and in vivo binding of COH2 to their promoters in Δ*Coh1* was also greater than in the WT strain (Fig 5F).

Using the protease gene *MAA_10199* as a representative of the 3 cuticle-degrading genes described above, the results about their expression obtained with surrogate hemocoel colonization were validated in the real hemocoel of infected insects by tracing GFP signal in the strains *WT-PMAA_10199-GFP* and Δ*Coh1-PMAA_10199-GFP*, which were constructed by transforming the *gfp* gene driven by the *MAA_10199* promoter into the WT strain and the mutant Δ*Coh1*, respectively (S1J Fig). The GFP fluorescent intensity in *WT-PMAA_10199-GFP* cells on the cuticle was stronger than in the real hemocoel (Fig 5G). In the real hemocoel, the GFP fluorescent intensity in the strain Δ*Coh1-PMAA_10199-GFP* was stronger than in *WT-PMAA_10199-GFP* (Fig 5G).

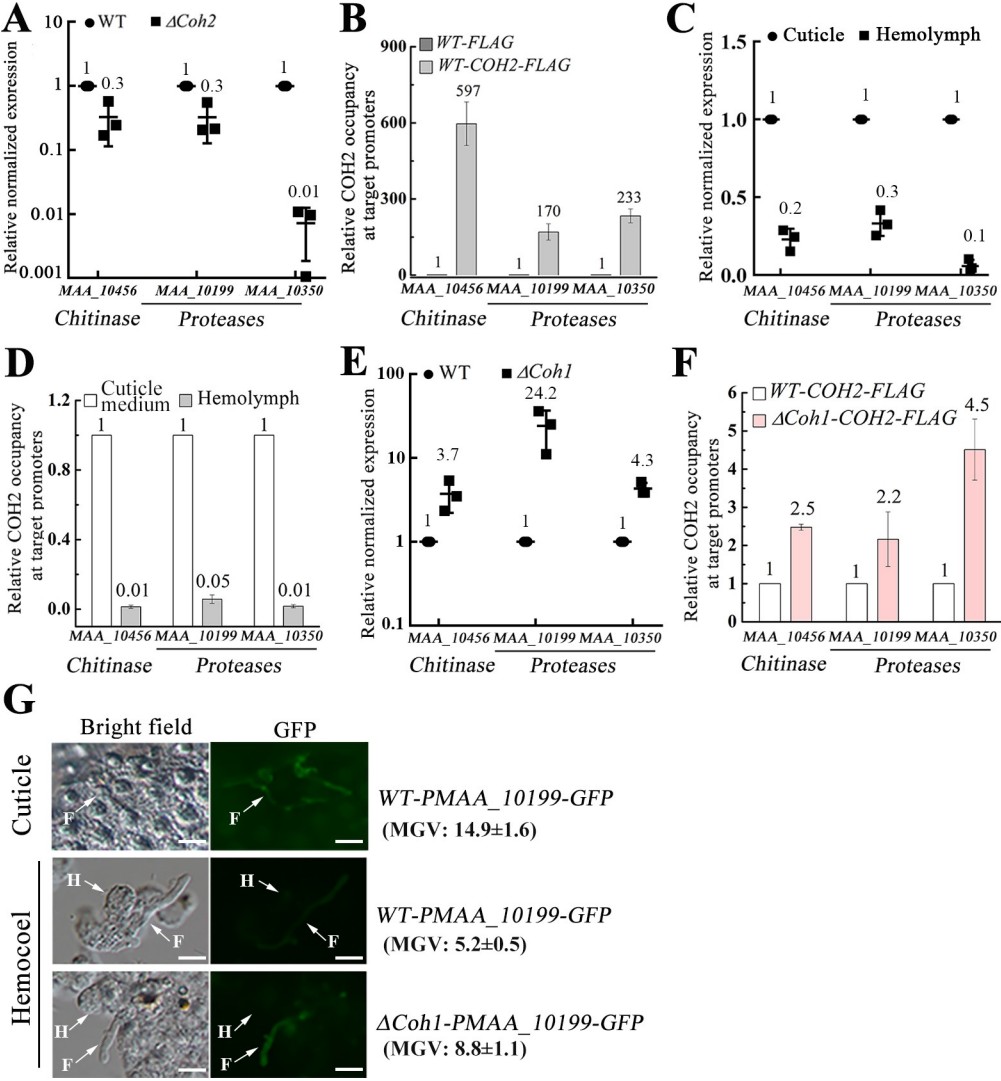

**Fig 5. COH1 inactivates the COH2-mediated induction of cuticle-degrading genes during hemocoel colonization.**
(A) Quantitative reverse transcription PCR (qRT-qPCR) analysis of the expression of cuticle-degrading genes during cuticle penetration in the mutant Δ*Coh2* relative to the wild-type (WT) strain. The chitinase MAA_10456 and the proteases MAA_10199 and MAA_10350 are used as representatives. (B) Chromatin immunoprecipitation quantitative PCR (ChIP-qPCR) analysis shows the occupancy of the COH2::FLAG protein in the promoters of the chitinase and protease genes in the strain *WT-COH2-FLAG* relative to occupancy in the control strain *WT-FLAG*, which is set to 1. The fungal strains were grown in the cuticle medium. (C) qRT-PCR analysis of the expression levels of the 3 genes during surrogate hemocoel colonization relative to the expression level during cuticle penetration, which is set to 1. (D) ChIP-qPCR analysis of the occupancy of COH2 in the promoters of the 3 genes during surrogate hemocoel colonization (Hemolymph) relative to the growth in the cuticle medium (Cuticle medium), which is set to 1. The strain *WT-COH2-FLAG* was used. (E) qRT-PCR analysis of the expression levels of the 3 genes during surrogate hemocoel colonization in the mutant Δ*Coh1* relative to the expression level in the WT strain, which is set to 1. (F) ChIP-qPCR analysis of the occupancy of the COH2::FLAG protein in the promoters of the 3 genes during surrogate hemocoel colonization in the strain Δ*Coh1-COH2-FLAG* (the mutant Δ*Coh1* with COH2::FLAG expressed) relative to occupancy in the strain *WT-COH2-FLAG*, which is set to 1. (G) GFP signal in the fungal cells of 2 strains (*WT-PMAA_10199-GFP* and Δ*Coh1-PMAA_10199-GFP*) with the *gfp* gene driven by the promoter of the protease gene *MAA_10199*. Top panel: *WT-PMAA_10199-GFP* on the cuticle (Cuticle); middle panel: *WT-PMAA_10199-GFP* in the real hemocoel of *G. mellonella* larva infected by a fungal strain (Hemocoel); bottom panel: Δ*Coh1-PMAA_10199-GFP* in the real hemocoel. F, fungal cells; H, hemocyte; MGV, mean gray value. Scare bar, 10 μm. The data underlying all the graphs shown in the figure can be found in S1 Data.

## Discussion

Fungi are the most common pathogens in insects, with approximately 1,000 species known to cause disease in arthropods [25]. During fungal pathogenesis in insects, fungi experience microenvironmental transition from the cuticle to the hemocoel. In this study, in the model entomopathogenic fungus *M. robertsii*, we discovered a novel regulatory cascade that controls the response of the fungus to the 2 microenvironments. In this cascade, the transcription factor COH2 interacts with the regulatory protein COH1 and acts as a switch to turn on or off genes involved in cuticle penetration or colonization of the hemocoel. During cuticle penetration, COH2 activates the expression of genes encoding cuticle-degrading proteases, chitinases, and lipases by directly binding to their promoter regions. In addition to the cuticle-degrading genes directly regulated by COH2, RNA-Seq analysis showed that other cuticle-degrading genes were indirectly regulated by COH2. *M. robertsii* has a complex cuticle-degrading enzyme system that contains many proteases, chitinases, lipases, and other enzymes, and cuticle degradation by some of these enzymes can induce the expression of additional cuticle degradation enzymes [26]. Therefore, the activities of the cuticle-degrading genes directly regulated by COH2 could induce the expression of other genes involved in cuticle penetration.

Conversely, the transcription factor COH2 represses the expression of genes for hemocoel colonization, including those for destruxin biosynthesis and for iron acquisition, when *M. robertsii* penetrates the cuticle and reaches the hemocoel. However, the Ecl1 domain-containing regulatory protein COH1 is expressed in the hemocoel and physically contacts COH2. This interaction suppresses COH2-mediated repression of genes for colonization of the hemocoel. Furthermore, the interaction between COH1 and COH2 inhibits COH2-mediated induction of cuticle-degrading genes to ensure that these genes are not expressed in the insect hemocoel. The precise regulation of these cuticle-degrading genes is important as they could, if expressed in the hemocoel, cause hyperactive immunity of the insect host [24]. The mechanism underlying how COH1 suppresses the activity of COH2 in the hemocoel is that their interaction renders COH2 susceptible to protein degradation by the 26S proteasome pathway. Although COH2 represses the expression of genes for hemocoel colonization, deleting the gene *Coh2*, i.e., completely removing COH2 protein, slightly but significantly impaired the ability of the fungus to colonize the insect hemocoel. A possible explanation for this discrepancy is that the 26S proteasome pathway does not completely degrade COH2 protein during hemocoel colonization and the remaining protein could have other uncharacterized functions to promote hemocoel colonization. In addition, RNA-Seq analysis showed that many genes were simultaneously regulated by COH1 and COH2, but COH1 and COH2 controlled a different set of other genes, suggesting that COH1 could also have other functions than interacting with COH2. This could explain why no difference in virulence was found between the mutants ΔCoh1, ΔCoh2, and ΔCoh1::ΔCoh2, and why the mutant ΔCoh2 differed from the Coh2-expressing strains ΔCoh1, ΔCoh1-COH2-FLAG, and WT-COH2-FLAG in the impact on host phenoloxidase activity. In several previous studies, Ecl1 proteins were described in the saprophytic yeasts *S. cerevisiae* and *Sch. pombe*, and they are considered to be regulators that are involved in chronological lifespan events [27], stress resistance [28], and sexual development [29], but their molecular functions remain unclear. We characterized the functions of the Ecl1 protein COH1 in the entomopathogenic fungus *M. robertsii*. Repression of *Coh1* by epigenetic regulation is important for saprophytic growth since the *Coh1*-overexpressing strain showed abnormal colony phenotype and reduced conidial yield. However, COH1 is expressed during hemocoel colonization, and plays important roles in this infection stage via interacting with the transcription factor COH2.

The expression of *Coh1* during hemocoel colonization is activated by reversion of the epigenetic repression conferred by the histone H3 deacetylase HDAC1 and the histone H3

acetyltransferase HAT1, which target H3K56 and H3K4, respectively. The reversion of the epigenetic repression by HDAC1 was confirmed by 2 lines of evidence. First, in the WT strain, the expression level of *Hdac1* during surrogate hemocoel colonization was significantly lower than during saprophytic growth. The acetylation level of H3K56 in the *Coh1* promoter during surrogate hemocoel colonization was higher than during saprophytic growth. Furthermore, during saprophytic growth, deleting the gene *Hdac1* resulted in an increase in the acetylation level of H3K56 in the *Coh1* promoter, thus activating *Coh1* expression. Second, during surrogate hemocoel colonization, deleting the gene *Hdac1* had no significant impact on the acetylation level of H3K56 in the *Coh1* promoter, but forcefully elevating *Hdac1* expression with a constitutive promoter decreased the acetylation level of H3K56 in the *Coh1* promoter and thus reduced the expression level of *Coh1*. The reason why deleting *Hdac1* did not result in significant increase in the acetylation level of H3K56 in the *Coh1* promoter during surrogate hemocoel colonization could be that *Hdac1* expression is reduced to a very low level, such that HDAC1 had negligible impact on the acetylation level of H3K56 in the *Coh1* promoter.

When *M. robertsii* reaches the insect hemocoel from the cuticle, the expression level of *Hat1* is reduced. This results in a decrease in the acetylation level of H3K4 in the *Hdac1* promoter and thus a reduction in the expression of this gene. In addition to regulation of the acetylation level of H3K56 in the *Coh1* promoter via controlling HDAC1, deleting *Hat1* also elevated the H3K4 acetylation level in the promoter of *Coh1*, suggesting that additional unknown components are involved in regulation of *Coh1* by HAT1. Reduction in *Hat1* expression in the hemocoel is not due to the feedback regulation by *Coh1* or *Coh2*, and is attributed to other uncharacterized factors. During the microenvironmental transition from the cuticle to the hemocoel, *M. robertsii* encounters vast environmental changes with respect to the host defense, nutrients, osmolarity, and availability of oxygen. The triggers for the transcriptional reprogramming controlled by the regulatory cascade could be multifactorial, which remains to be resolved.

In conclusion, we discovered a regulatory cascade that precisely controls different sets of genes for the responses of *M. robertsii* to 2 distinctive microenvironments (the cuticle and the hemocoel) during its pathogenesis in insects. This cascade contains 4 components: the histone 3 acetyltransferase HAT1, the histone 3 deacetylase HDAC1, the transcription factor COH2, and its interacting protein COH1. On the insect cuticle, COH2 switches on key genes for cuticle penetration. Once the fungus reaches the hemocoel, the expression of 2 epigenetic regulators HAT1 and HDAC1 is reduced, and this increases the acetylation level of histone H3 in the *Coh1* promoter and thus activates its expression. COH1 physically contacts COH2 to reduce COH2 stability, which thus switches off genes for cuticle penetration and switches on key genes for colonization of the hemocoel. Our work significantly advances the insights into fungal pathogenesis in insects. Homologs of the 4 regulatory genes described in *M. robertsii* are widely found in other fungi including insect pathogenic fungi such as *Beauveria bassiana* and human pathogenic fungi such as *Talaromyces marneffei*. Therefore, similar regulatory cascades could exist in many other pathogenic fungi.

## Methods

### Gene deletion

*M. robertsii* ARSEF2575 was obtained from the Agricultural Research Service Collection of Entomopathogenic Fungal Cultures (US Department of Agriculture). *E. coli* DH5α was used for constructing plasmids. Transformations of *M. robertsii* were mediated by *Agrobacterium tumefaciens* AGL1 [30].

Gene deletion and complementation of gene deletion mutants were conducted as previously described [30]. The plasmids for gene deletion were constructed using pPK2-Bar-GFP or

pPK2-Sur-GFP as master plasmids [30]. To construct a plasmid for complementing a gene deletion mutant, the genomic clone of the gene—containing the promoter region, ORF, and termination region—was amplified by PCR and inserted into the plasmid pPK2-Sur-GFP [30]. High-fidelity Taq DNA polymerase (KOD Plus Neo; Toyobo, Osaka, Japan) was used in PCR reactions, and all PCR products were confirmed by DNA sequencing. The primers for the construction of gene deletion plasmids and other primers used in this study are presented in S1 Table.

### Construction of strains overexpressing *Coh1 or Hdac1*

The coding sequences of *Coh1* and *Hdac1* were cloned by PCR and inserted downstream of the constitutive promoter *Ptef* in the plasmid pPK2-Bar-Ptef [2] to produce plasmid pPK2-Bar-Ptef-Coh1 and pPK2-Bar-Ptef-Hdac1, respectively. The resulting plasmids were then transformed into the WT strain to produce the strains $Coh1^{OE}$ and $Hdac1^{OE}$. The overexpression of the 2 genes was confirmed by qRT-PCR.

### Assays of phenoloxidase activity and antimicrobial expression

To assay the impact of fungal infection on phenoloxidase activity and expression of antimicrobial-encoding genes in the *G. mellonella* larvae, 5 μL of a conidial suspension ($1 \times 10^7$ conidia/mL) was injected into each larva. After 12 h, the hemolymph of 3 larvae was collected into PBS (phosphate buffered saline) buffer at 4°C, and the protein concentration was determined using the BCA Protein Assay Kit (Meilune, Dalian, China). The hemolymph (5 μg of total protein) was suspended in a $CaCl_2$ solution (5 mM) to achieve a final volume of 20 μl, which was then mixed with 80 μL of L-DOPA (L-3,4-dihydroxyphenylalanine) solution (20 mM, pH 6.6). After 30 min of incubation at 26°C in the dark, the absorbance at 492 nm was measured. Phenoloxidase activity in the hemolymph was expressed as the absorbance per microgram of protein. L-DOPA solution was used as the blank [31,32]. The control was the hemolymph collected from the uninoculated insects. This experiment was repeated 3 times.

qRT-PCR analysis of the expression levels of antimicrobial-encoding genes was conducted with RNA prepared from the fat bodies of the infected *G. mellonella* larvae. The *G. mellonella* 18S rRNA gene (GenBank accession number: AF286298) was used as an internal standard.

### Analysis of fungal burden in insects

Absolute qPCR analysis was used to assay the fungal burden in the insects infected by *M. robertsii* by quantifying the fungal genomic DNA. The 18s rDNA internal transcribed spacer (ITS) sequence was used for qPCR analysis to estimate fungal burden. To construct a standard curve for the absolute qPCR analysis, the genomic DNA of *M. robertsii* was extracted from mycelium grown in SDY medium, and a serial dilution was made as templates for qPCR. To extract DNA from the infected insects, 5 larvae were surface sterilized with sodium hypochlorite solution (1%) and pooled for DNA extraction. The fungal burden was expressed as the amount of fungal genomic DNA per 1 gram of infected larvae.

### Co-IP analysis

In addition to the master plasmids pPK2-Sur-Ptef-FLAG, pPK2-Sur-Ptef-HA, pPK2-Bar-Ptef-FLAG, and pPK2-Bar-Ptef-HA that were used for producing a protein fused with FLAG or HA [33], in this study we also constructed the master plasmid pPK2-Bar-Ptef-Myc by inserting the coding sequence corresponding to 13 repeats of the Myc peptide in the plasmid pPK2-Bar-Ptef [2]. All fusion genes were driven by the constitutive promoter *Ptef* from *A. pullulans* [18].

To assay the in vivo physical interaction of COH1 with COH2, the strain *COH1-HA/COH2-Myc* simultaneously expressing the fusion proteins COH1::HA and COH2::Myc was constructed. The coding sequence of COH1 (stop codon excluded) was cloned by PCR and inserted upstream of the coding sequence of the HA tag in the plasmid pPK2-Sur-Ptef-HA to produce the plasmid pPK2-Sur-Ptef-COH1-HA, which was then transformed into the WT strain to produce a strain called *COH1-HA*. The coding sequence of COH2 (stop codon excluded) was also amplified by PCR and inserted upstream of the coding sequence of the Myc tag in the plasmid pPK2-Bar-Ptef-Myc, producing the plasmid pPK2-Bar-Ptef-COH2-Myc. This plasmid was then transformed into the strain *COH1-HA* to produce the strain *COH1-HA/COH2-Myc* for Co-IP analysis. Similarly, to identify the section of COH2 that physically contacted COH1, the protein was divided into 2 portions: the N-terminus (Met-1 to Thr-100) containing the DNA binding domain COH2-DBD and NLS, and the C-terminus (Ser-101 to Arg-282). The coding sequences of the N-terminus and C-terminus were cloned by PCR and inserted upstream of the coding sequence of the Myc tag in the plasmid pPK2-Bar-Ptef-Myc. The resulting plasmids were then transformed into the strain *Coh1-HA* to produce the strains *COH1-HA/COH2-N-Myc* and *COH1-HA/COH2-C-Myc* for Co-IP analysis.

To assay whether COH2 formed a homodimer, the strain *COH2-FLAG/COH2-Myc* was constructed. To do this, the plasmid pPK2-Bar-Ptef-COH2-Myc was transformed into the WT strain to produce a strain called *COH2-Myc*. The coding sequence of COH2 (stop codon excluded) was amplified by PCR and inserted upstream of the coding sequence of the FLAG tag in pPK2-Sur-Ptef-FLAG, producing the plasmid pPK2-Sur-Ptef-COH2-FLAG. This plasmid was then transformed into the strain *COH2-Myc* to produce the strain *COH2-FLAG/COH2-Myc* for Co-IP analysis. To assay the impact of COH1 on COH2 dimerization, the coding sequence of COH1 was cloned and inserted downstream of the promoter *Ptef* in the plasmid pPK2-NTC-GFP-Ptef [26] to produce the plasmid pPK2-NTC-GFP-Ptef-COH1, which was then transformed into the strain *COH2-FLAG/COH2-Myc* to produce the strain *Coh1$^{OE}$-COH2-FLAG/COH2-Myc*.

Co-IP analysis was performed as previously described [33]. Protein concentration was determined using the BCA Protein Assay Kit (Meilune, Dalian, China). The mouse anti-Myc and anti-HA antibodies were purchased from ABclonal (Wuhan, China), and the rabbit anti-FLAG antibody was purchased from Sigma-Aldrich (St. Louis, MO, US). The Dynabeads protein G beads were purchased from Invitrogen (Carlsbad, CA, US).

Confirming the expression of the fusion proteins in the fungal strains and assaying the presence of target proteins after Co-IP were conducted using standard Western blot analysis. Total fungal proteins were extracted as previously described [33]. All Co-IP assays were repeated at least 3 times.

## ChIP-Seq and ChIP-qPCR analysis

The strain *WT-COH2-FLAG* expressing the fusion protein COH2::FLAG was used for ChIP-Seq to identify the DNA motif bound by the transcription factor COH2. Conidia ($10^8$) were inoculated in 100 mL of SDY medium and incubated at 26°C for 36 h with 220 rpm shaking. The mycelium was then collected and subjected to ChIP-Seq analysis that was conducted by the company Wuhan Igenebook Biotechnology (Wuhan, China). The antibody used for immunoprecipitation was anti-FLAG (Huabio, Hangzhou, China), and ChIP-enriched DNA was sequenced using the Illumina HiSeq 2000 sequencing system. After initial quality control to remove sequencing adaptors and low-quality bases, the clean reads were mapped to the genome of *M. robertsii* using the software BWA (version 0.7.15-r1140). The peak caller MACS

(version 2.1.1.20160309; *q* value < 0.05) was used to localize the potential binding sites of COH2 [34].

ChIP-qPCR was performed as previously described [33], with some modifications. The cross-linked mycelium was ground into fine powder in liquid nitrogen and then resuspended in a lysis buffer. After centrifugation, the chromatin DNA was sheared using M220 (Covaris, Woburn, MA, US) with PIP 75 W, duty factor 10, cycles/burst 200 count, and time 1,500 s. Immunoprecipitation was performed using antibodies (anti-FLAG [Sigma-Aldrich, St. Louis, MO, US], anti-H3Ac, anti-H3K4Ac, or anti-H3K56Ac [Active Motif, Carlsbad, CA, US]) together with Dynabeads protein G beads (Invitrogen, Carlsbad, CA, US). ChIP-enriched DNA was used as a template for qPCR analysis using Thunderbird SYBR qPCR Mix without ROX (Toyobo, Osaka, Japan). Relative enrichment values were calculated by dividing the immunoprecipitated DNA by the input DNA. All ChIP-qPCR analyses were repeated 3 times.

To prepare the mycelium during the surrogate hemocoel colonization, the hemolymph (250 μL) was collected from the last instar *G. mellonella* larvae and mixed with an equal volume of anticoagulant solution (60 mM trisodium citrate, 52 mM citric acid, 120 mM NaCl, 200 mM glucose, 20 mM EDTA), into which 10 μL of a conidial suspension ($1 \times 10^7$ conidia/mL) was inoculated. After incubation at 26˚C for 36 h with gentle shaking, the culture was treated with a cycle of freezing (−80˚C) and thawing (26˚C), and then subjected to centrifugation. The pellet was resuspended in sterile water and collected again with centrifugation. This washing with water was repeated 7 times to remove insect hemolymph, and the mycelium was then collected for ChIP-qPCR analysis. To obtain enough biomass to get sufficient DNA for ChIP-qPCR analysis during cuticle penetration, fungal mycelium was grown in the cuticle medium with the cuticle of *G. mellonella* larvae (1%, V/W) as the sole carbon and nitrogen source. Preparation of the *G. mellonella* cuticle was conducted as previously described [26]. To this end, the mycelium grown in the SDY medium described above was collected by filtration and then washed with sterile water 3 times. The mycelium was then grown at 26˚C for 1 h in the cuticle medium, then collected for ChIP-qPCR analysis.

## EMSAs

EMSAs were conducted as previously described [33] to analyze the binding of the COH2 DNA binding domain (COH2-DBD) to a DNA probe containing *COH2-BM*. The DNA probe was a portion of the promoter of the gene MAA_04430, which was shown to be bound by COH2 by the ChIP-Seq analysis. To express COH2-DBD (Ser-30 to Thr-100) in *E. coli*, its coding sequence was cloned by PCR and inserted into the EcoRI/HindIII sites of the plasmid pET28a (Invitrogen). The resulting plasmid was transferred into the *E. coli* BL-21 strain, and the expression of the recombinant protein was induced by isopropyl β-D-1-thiogalactopyranoside at 18˚C for 16 h as described in the manufacturer's instructions (Novagen, Madison, WI, US). The recombinant COH2-DBD protein was then purified to homogeneity with HisPur Ni-NTA resin (Thermo Fisher Scientific, Waltham, MA, US) and gel filtration chromatography with HiLoad 16/600 and 26/600 Superdex 75 prep grade (GE Healthcare Bio-Sciences, Uppsala, Sweden).

To prepare the DNA probe labeled with biotin, one of its strands was biotin-labeled (Tsingke Biological Technology, Hangzhou, China), which was annealed to its complementary strand to form double-strand DNA according to the manufacturer's instructions. Two unlabeled strands of the DNA probe were commercially synthesized and annealed to form the unlabeled DNA fragment. A mutated DNA probe was prepared exactly as the WT DNA probe with 1 nucleotide in the motif *COH2-BM* substituted to each of the other 3 ones. The details of the mutated DNA probes are shown in S10E Fig.

EMSAs were conducted using the LightShift Chemiluminescent EMSA Kit (Thermo Fisher Scientific). In the competition assays, an unlabeled DNA probe was added in 300-fold excess.

## RNA-Seq and qRT-PCR analysis

Total RNA was extracted with the TRIzol reagent (Thermo Fisher Scientific). The mycelium during saprophytic growth and surrogate hemocoel colonization was prepared as described above for ChIP-qPCR analysis, and was then subjected to RNA extraction. To prepare the mycelium during cuticle penetration, a conidial suspension was applied to locust hindwings placed on a water agar plate. After 30 h of incubation at 26°C, the hindwings and fungal biomass were subjected to RNA extraction.

RNA-Seq analysis was performed by Personal Gene Technology (Nanjing, China). Paired-end sequencing was conducted on the Illumina HiSeq 2000 sequencing platform. Clean reads were mapped to the *M. robertsii* genome using software HISAT2 (https://daehwankimlab.github.io/hisat2/). Reads that aligned uniquely to the reference sequence were used for gene expression quantification using the FPKM (fragments per kilobase per million fragments) method. Differential expression analysis was performed with DESeq software with an adjusted *P* value 0.05 (Benjamini–Hochberg method). For the RNA collected from the mycelium grown in the hemolymph, the reads that originated from the insect RNA were filtered, and the remaining reads were mapped to the *M. robertsii* genome for subsequent analyses.

For qRT-PCR, complementary DNAs (cDNAs) were synthesized with total RNAs with ReverTra Ace qPCR RT Master Mix (Toyobo, Osaka, Japan). qRT-PCR analysis was conducted using Thunderbird SYBR qPCR Mix without ROX (Toyobo, Osaka, Japan). The genes *Gpd* and *tef* were used as internal standards as previously described [35]. The relative expression level of a gene was determined using the $2^{-\Delta\Delta Ct}$ method [36]. All qRT-PCR experiments were repeated 3 times.

## Assay of the acetylation level in histone H3

To analyze the acetylation level of histone H3, 100 μg of total protein was electrophoresed on a 12% SDS-PAGE gel and then transferred to a PVDF membrane (Bio-Rad, Hercules, CA, US). The global acetylation level of histone H3 was detected by anti-acetyl histone H3 (Sigma-Aldrich). Antibodies against the acetylation levels of different lysine sites in histone H3 (H3K4, H3K9, H3K14, H3K18, H3K23, H3K27, and H3K56) were all purchased from Active Motif (Carlsbad, CA, US) and diluted 1:2,000 for Western blot analysis. Histone H3 protein was assayed with anti-H3 (Merck Millipore; 1:1,000 dilution). All blots were imaged by the chemiluminescence detection system (Clarity Western ECL, Bio-Rad), and the signal intensities of all bands were quantified using Image J software (https://imagej.nih.gov/ij). The experiments were repeated 3 times.

## Yeast 2-hybrid and autoactivation assays

Yeast 2-hybrid assays were performed according to the manufacturer's instructions (Clontech, Japan). The interactions between COH1 and 5 transcription factors were assayed. The coding sequence of each of the 5 transcription factors (MAA_08013, MAA_06777, MAA_07566, MAA_06937, and MAA_07838) was cloned by PCR and inserted into the plasmid pGADT7. The coding sequence of COH1 was also cloned by PCR and inserted into the plasmid pGBKT7 to produce the plasmid pGBKT7-COH1. All pGADT7-based plasmids were transformed into Y187 cells, whereas pGBKT7-COH1 was transformed into Y2HGold cells. The Yeastmaker Yeast Transformation System 2 (Takara Bio, Tokyo, Japan) was used to prepare yeast-competent cells. After mating, the resulting strains were grown on the medium SD/−Ade/−His/

−Leu/−Trp with X-α-gal and AbA (Aureobasidin A; Takara Bio). The autoactivation of COH1 was tested by inoculating the strain containing the plasmid pGBKT7-COH1 on the medium SD/−Ade/−His/−Trp with X-α-gal. Yeast 2-hybrid and autoactivation assays were repeated 3 times.

## Inhibition of the activity of the ubiquitin–proteasome pathway

Conidia ($10^8$ in total) were inoculated into 100 mL of SDY medium and grown at 26˚C for 30 h, and the 26S proteasome inhibitor MG132 (200 µM) was then added. After 4 h of incubation, the mycelium was collected by filtration for protein preparation. To assay the ubiquitination level of the protein COH2::FLAG, immunoprecipitation with the mouse anti-FLAG (HUA-BIO, China) was performed as previously described [30]. The rabbit anti-ubiquitin antibodies were purchased from ABclonal (Wuhan, China).

## Histoimmnunochemical staining

Histoimmnunochemical staining analysis of the fusion protein COH2::FLAG was conducted as previously described [2]. The primary antibody was anti-FLAG (HUABIO, China), and fluorescein isothiocyanate (FITC)–conjugated goat anti-mouse IgG (HUABIO, China; dilution 1:200) was used for secondary labeling.

## Analysis of gene expression by tracing GFP signal

To investigate expression of a gene in the real hemocoel of the infected insects by tracing GFP signal, we constructed a strain with GFP coding sequence driven by the promoter of this gene. The conidia ($10^5$ conidia in 10 µL of 0.01% Triton X-100 solution) of this fungal strain were injected into a *G. mellonella* larva. After 36 h, the fungal cells in the hemocoel were collected for GFP signal observation and quantification. To analyze the expression of the gene during cuticle penetration, 200 µL of the conidial suspension was inoculated on the *G. mellonella* cuticle, and GFP signal was observed and quantified after 36 h of incubation.

The fluorescence intensity across a hypha in the images described above was quantified using ImageJ software (https://imagej.nih.gov/ij). To this end, an image was saved in a Tiff format, and its bit depth was then changed to 8 bits, followed by subtracting the general background using a 50-pixel "rolling ball diameter" [37]. The hyphae in the image were then selected, and the area, the mean gray value (MGV), and the MGV's standard deviation for the selected regions were then calculated. The fluorescence intensity of the selected hyphae is shown as the MGV.

## Pathogenicity assays

The *G. mellonella* larvae (RuiQing Bait Company, Shanghai, China) were used to assay the pathogenicity of *M. robertsii*. Inoculations were conducted via topical application of conidia on the insect cuticle or by injection of conidia into the insect hemocoel [10]. All bioassays were repeated 3 times with 40 insects per repeat. Assays of appressorial formation on the hydrophobic surface of a petri dish (Corning, Corning, NY, US) were conducted as previously described [2].

The ability to penetrate the insect cuticle was assayed as previously described [33]. Briefly, 2 µL of a conidial suspension ($1 \times 10^7$ conidia/mL) was applied on the center of an intact cuticle that was placed on a PDA plate. After incubation at 26˚C for 48 h, the cuticle was removed, and the PDA plate was further incubated to allow the growth of the fungi, if any, that had breached the cuticle and reached the PDA medium. The stronger the ability of the fungi to

penetrate the insect cuticle, the more fungi reached the PDA medium, i.e., the larger the colony that appeared on the PDA plate following removal of the cuticle and incubation.

To assay the ability of the fungus to produce blastospores in the insect hemocoel, 5 μL of a conidial suspension ($1 \times 10^5$ conidia/mL) was injected into the hemocoel of *G. mellonella* larvae. Then, 3 and 4 d after injection, 10 larvae per treatment were surface sterilized with 1% bleach and individually bled as previously described [10]. Aliquots of hemolymph (50 μL) from each larva were mixed with 50 μL of the anticoagulant solution and spread onto a selective medium [10]. After incubation at 26˚C for 4 d, the colony forming units (CFU) were determined. All assays were repeated at least 3 times.

## Supporting information

**S1 Data. Underlying numerical data for each figure.**
(XLSX)

**S1 Fig. Confirmation of deletion of the gene *Coh1*, *Hdac1*, or *Coh2*, and overexpression of *Hdac1* and *Coh1*.** (A) The schematic diagram of gene deletion based on homologous recombination. Lower panel is a map of a deletion plasmid, and its relative position in the fungal genome is shown in the upper panel. (B) Confirmation of construction of the *Coh1* deletion mutant (Δ*Coh1*) using PCR in the transformants with herbicide PPT resistance and without a GFP signal. Δ*Coh1#1*, Δ*Coh1#2*, and Δ*Coh1#3* represent 3 independent isolates of the deletion mutant, and WT is the wild-type strain. Upper panel: PCR conducted with the primers Bar-up and the confirmation primer CF2 (the relative positions of all primers are shown in [A]). PCR products can be obtained only from the deletion mutants. Lower panel: PCR conducted with primers CF1 and CF2. PCR products can be obtained only in the WT strain. (C) Southern blot analysis confirms that the selection marker gene *Bar* was not ectopically integrated in the 3 isolates of the *Coh1* deletion mutant shown in (B), indicating that the insertion of the selection marker gene only deleted the *Coh1* gene. Genomic DNA (15 μg) was digested with BamHI and SmaI. The PCR product of the *Bar* gene was digoxigenin (DIG)–labeled and used as the DNA probe. (D) Confirmation of the insertion of the genomic clone of the *Coh1* gene into the genome of the deletion mutant Δ*Coh1* for the complementation by PCR using the primers ORF-5 and ORF-3. C1, C2, C3, C4, and C5 represent 5 randomly selected transformants. (E) Reverse transcription PCR (RT-PCR) analysis of *Coh1* expression in the 5 randomly selected transformants (C1, C2, C3, C4, and C5) shown in (D), suggesting that construction of the complemented strain of Δ*Coh1* failed due to the importance of the native chromosomal position of *Coh1* for its transcriptional regulation. RNA was collected from the mycelium grown on PDA plates, where *Coh1* was not expressed in the WT strain. Upper panel: the *Coh1* gene. Note: No PCR product was seen in the WT strain and the deletion mutant Δ*Coh1*. Lower panel: the reference gene *Gpd* encoding glyceraldehyde 3-phosphate dehydrogenase. (F) RT-PCR confirmation of *Coh1*-overexpressing strain *Coh1*$^{OE}$ during saprophytic growth. *Coh1*$^{OE}$*#1*, *Coh1*$^{OE}$*#2*, and *Coh1*$^{OE}$*#3* are 3 independent isolates of the strain *Coh1*$^{OE}$. The isolate *Coh1*$^{OE}$*#3* was used for construction of other strains. (G) PCR confirmation of construction of the deletion mutants Δ*Hdac1* (left) and Δ*Coh2* (middle) and the double deletion mutant Δ*Coh1*::Δ*Coh2* (right). To construct the mutant Δ*Coh1*::Δ*Coh2*, the *Coh1* gene was deleted in the mutant Δ*Coh2*. For each gene, the legends for the upper and lower panels are the same as in (B). D1, D2, and D3 represent 3 independent deletion mutants. (H) PCR confirmation of the complementation of the deletion mutants Δ*Hdac1* and Δ*Coh2* with their respective genomic clones. C1 and C2 represent 2 independent complemented strains. (I) qRT-PCR confirmation of construction of the *Hdac1*-overexpressing strain *Hdac1*$^{OE}$ or Δ*Hat1-Hdac1*$^{OE}$ during surrogate hemocoel colonization. All qRT-PCR analyses in this figure were repeated 3

times, and the values represent the fold-change of expression of a gene in treatment compared with expression in its respective control, which is set to 1. (J) PCR confirmation of construction of strains expressing GFP driven by different promoters. The data underlying the graph shown in (I) can be found in S1 Data.
(TIF)

**S2 Fig.** *Coh1* **is involved in hemocoel colonization but not in cuticle penetration, saprophytic growth, and conidiation.** (A) The colony diameter on PDA plates. ΔCoh1#1, ΔCoh1#2, and ΔCoh1#3 are 3 independent isolates of the mutant ΔCoh1. Note: At each time point after inoculation, no significant difference in colony diameter was found between all tested strains ($n = 3$, $P > 0.05$, Tukey's test in one-way ANOVA). (B) Conidial yields on PDA plates. Quantification of conidial yield was repeated 3 times with 3 replicates per repeat. Data are expressed as mean ± SE. No significant difference in conidial yield was found between the strains ($n = 9$, $P > 0.05$, one-way ANOVA). (C) Colony phenotypes (upper panel) and conidiophores (lower panel) on PDA plates of the 4 strains described in (A). Conidiophores were observed 5 d after inoculation (scale bar represents 10 μm), while colony picture was taken 18 d after inoculation (scale bar represents 10 mm). Images are representative of at least 3 independent experiments. (D) $LT_{50}$ (time taken to kill 50% of insects) values when the insects were inoculated by injection of conidia into the hemocoel. Data are expressed as mean ± SE. Values with different letters are significantly different ($n = 3$, $P < 0.05$, Tukey's test in one-way ANOVA). (E) Quantification of hyphal bodies in the insect hemocoel at day 2, 3, and 4 after injection of conidia into the hemocoel. CFU, colony forming units. Within each day, values with different letters are significantly different ($n = 3$, $P < 0.05$, Tukey's test in one-way ANOVA). (F and G) Absolute qPCR analysis of fungal burden in live insects infected by *M. robertsii* strains via (F) topical application or (G) direct injection. Values with different letters are significantly different ($n = 5$, $P < 0.05$, Tukey's test in one-way ANOVA). (H) Appressorial development on a hydrophobic plastic surface. For all time points, no significant difference in appressorial formation was found between the WT strain and the 3 independent isolates of the mutant ΔCoh1 ($n = 3$, $P > 0.05$, Tukey's test in one-way ANOVA). The experiment was repeated 3 times. Data are expressed as mean ± SE. The insets are representatives of appressoria formed at 16 h after inoculation. (I) Assays of penetration of the cuticle of *G. mellonella* larvae. Conidia (2 μl of a conidial suspension [$1 \times 10^7$ conidia/mL]) were applied on the cuticle, which was placed on a PDA plate and incubated at 26°C for 48 h to allow penetration. The cuticles were removed, and the plates were then incubated at 26°C to allow fungal growth. Scale bar represents 1 cm. The appearance of fungal colonies indicates the success of cuticle penetration. The size of the fungal colony on the PDA plate represents the ability to penetrate the cuticle. Images are representative of 3 independent experiments. (J) Phenoloxidase activity in the hemolymph collected from *G. mellonella* larvae infected by different *M. robertsii* strains. Control: uninoculated insects. Values with different letters are significantly different ($P < 0.05$, Tukey's test in one-way ANOVA). (K) qRT-PCR analysis of the expression levels of the genes encoding antimicrobial *gallerimycin* and *defensin* in *G. mellonella* larvae. Control: uninoculated insects. Values with different letters are significantly different ($P < 0.05$, Tukey's test in one-way ANOVA). The data underlying all the graphs shown in this figure can be found in S1 Data.
(TIF)

**S3 Fig. Saprophytic growth and pathogenicity of the** *Coh1*-**overexpressing strain** *Coh1*$^{OE}$ **and the complemented strain** *C-ΔCoh1* **of** *ΔCoh1*. (A) Colony growth on PDA plates. *Coh1*$^{OE}$#1 and *Coh1*$^{OE}$#2 are 2 independent isolates of the strain *Coh1*$^{OE}$. No significant difference was found between the strains at each time point ($n = 9$, $P > 0.05$, one-way ANOVA). (B) Conidial yield on PDA plates. Values with different letters are significantly different ($n = 9$,

$P < 0.05$, Tukey's test in one-way ANOVA). (C) Conidiophores (upper panel; scale bar: 10 μm) and colony morphology (lower panel; scale bar: 1 cm) on PDA plates. (D and E) $LT_{50}$ values via (D) topical application or (E) direct injection. No significant difference was found between tested strains ($P > 0.05$, Tukey's test in one-way ANOVA). (F) Cuticle penetration. The scale bar represents 1 cm. Images are representative of 3 independent experiments. The data underlying all the graphs shown in this figure can be found in S1 Data.
(TIF)

**S4 Fig. *Coh1* expression in 9 epigenetic mutants.** (A) Reverse transcription PCR (RT-PCR) analysis of *Coh1* expression in the WT strain and the deletion mutants of 2 histone acetyltransferase genes (*Hat2* [*MAA_04679*] and *Hat3 MAA_04734*]), 5 histone deacetylase genes (*Hdac2* [*MAA_02065*], *Hdac3* [*MAA_04246*], *Hdac4* [*MAA_04846*], *Hdac5* [*MAA_05326*], and *Hdac6* [*MAA_05985*]), and 2 histone methyltransferase genes (*Hmt1* [*MAA_00272*] and *Hmt2* [*MAA_09293*]) during saprophytic growth. Note: No RT-PCR product was seen in the WT strain. Upper panel: the *Coh1* gene; lower panel: the reference gene *Gpd*. M, DNA ladder. Images are representative of 3 independent experiments. (B) qRT-PCR analysis of the expression of the 9 epigenetic genes described in (A) during surrogate hemocoel colonization (Hemolymph) and cuticle penetration (Cuticle) relative to saprophytic growth (SDY). (C) qRT-PCR analysis of *Coh1* expression in the strains Δ*Hat1* and Δ*Hat1-Hdac1*$^{OE}$ during surrogate hemocoel colonization. The data underlying all the graphs shown in this figure can be found in S1 Data.
(TIF)

**S5 Fig. Western blot analysis of expression of fusion proteins in different strains.** (A) Expression of the fusion protein COH2::FLAG in the WT strain to produce the strain *WT-COH2-FLAG*, in the strain *Coh1*$^{OE}$ to generate *Coh1*$^{OE}$*-COH2-FLAG*, and in the mutant Δ*Coh1* to form Δ*Coh1-COH2-FLAG*. The strain *WT-FLAG* was obtained by transforming the WT strain with the plasmid pPK2-Bar-Ptef-FLAG for the expression of the FLAG tag. Note: Due to dimerization, the band corresponding to the fusion protein COH2::FLAG was approximately 2-fold bigger than its predicted molecular weight. The band corresponding to the FLAG tag was not seen due to its small size. (B) Expression of the fusion protein COH1::HA in the WT strain to produce the strain *WT-COH1-HA*. The strain *WT-HA* was the WT strain transformed with the plasmid pPK2-Sur-Ptef-HA for expressing the HA tag. (C) Expression of the fusion protein COH2-N::GFP with the N-terminus (Met-1 to Thr-100) of COH2 tagged with the GFP protein in the WT strain to produce the strain *WT-COH2-N-GFP* and in the strain *Coh1*$^{OE}$ to produce *Coh1*$^{OE}$*-COH2-N-GFP*. *Coh1*$^{OE}$*-COH2-N-GFP#1*, *Coh1*$^{OE}$*-COH2-N-GFP#2*, and *Coh1*$^{OE}$*-COH2-N-GFP*#3 are 3 independent isolates of the strain *Coh1*$^{OE}$*-COH2-N-GFP*. (D) Expression of the fusion protein COH2::Myc in the strain *WT-COH2-FLAG* to produce the strain *COH2-FLAG/COH2-Myc*. The strain *WT-COH2-FLAG/Myc* is the strain *WT-COH2-FLAG* transformed with the plasmid pPK2-Bar-Ptef-Myc for expressing the Myc tag. (E) Expression of the protein with the tag Myc fused to the mutated COH2 (COH2$^{ΔDID}$, substituting the 7 leucine residues in the dimer interface domain with alanine) in the WT strain to produce the strain *WT-COH2*$^{ΔDID}$*-Myc*. The strain *WT-Myc* is the WT strain transformed with the plasmid pPK2-Bar-Ptef-Myc for expressing the Myc tag. (F) Expression of the fusion protein COH1::HA and the protein with the N-terminus of COH2 fused to the Myc tag in the WT strain to produce the strain *COH1-HA/COH2-N-Myc*. The strain *COH1-HA/Myc* is obtained by transforming the plasmid pPK2-Bar-Ptef-Myc into the strain *WT-COH1-HA*. (G) Expression of the fusion protein COH1::HA and the protein with the C-terminus of COH2 (Ser-100 to Arg-282) fused to the Myc tag in the WT strain to produce the strain *COH1-HA/COH2-C-Myc*. The strain *COH1-HA/Myc* is described in (F). The

strain *WT-Myc* is described in (E). The antibody used is shown at the bottom of each panel. All images are not cropped. Images of all Western blot analyses are representative of 3 independent experiments for each condition.
(TIF)

**S6 Fig. Confirmation of COH2 dimerization.** (A) Co-IP confirmation of the formation of COH2 dimer. The fusion proteins COH2::FLAG (molecular weight = 33 kDa) and COH2::Myc (molecular weight = 44.6 kDa) were simultaneously expressed in the strain *COH2-FLAG/COH2-Myc*. The control strain *COH2-Myc* expressed the protein COH2::Myc. Immunoprecipitation was conducted with an anti-FLAG antibody. Proteins were detected by immunoblot analysis with anti-Myc or anti-FLAG antibodies. Upper panel: arrow indicates the dimer of COH2::FLAG; lower panel: arrow indicates the dimer of COH2::Myc and COH2::FLAG. (B) The leucine residues in the dimer interface domain (DID) are key to COH2 dimerization. *WT-COH2-Myc*: a strain expressing the protein COH2::Myc; *WT-COH2$^{ΔDID}$-Myc*: a strain expressing the Myc-tagged protein COH2$^{ΔDID}$, a mutated COH2 with the 7 leucine residues in the DID substituted to alanine. Note: The protein COH2::Myc forms a dimer (upper panel), whereas COH2$^{ΔDID}$::Myc does not (lower panel). (C) Co-IP analysis shows that the Myc-tagged N-terminus of COH2 (COH2-N::Myc) physically interacts with the HA-tagged COH1 (COH1::HA). (D) Co-IP analysis shows that the Myc-tagged C-terminus of COH2 (COH2-C::Myc) does not interact with the HA-tagged COH1 (COH1::HA). Immunoprecipitation was conducted with anti-HA antibody. Proteins were detected by immunoblot analysis with anti-HA or anti-Myc antibodies.
(TIF)

**S7 Fig. The fusion protein COH2::FLAG has the same functions as COH2.** (A) Colony diameters of the WT, *ΔCoh2-PCoh2-COH2-FLAG*, and *C-ΔCoh2* strains. Note: There was no significant difference between the 3 strains within each day ($n = 9$, $P > 0.05$, Tukey's test in one-way ANOVA). (B) LT$_{50}$ values via topical application. No significant difference was found between the tested strains ($n = 3$, $P > 0.05$, Tukey's test in one-way ANOVA). The data underlying all the graphs shown in this figure can be found in S1 Data.
(TIF)

**S8 Fig. Impacts of COH1 and COH2 interaction on COH2 stability, COH2 dimerization, and fungus entry into the nucleus.** (A) qRT-PCR analysis of the expression of the gene encoding the fusion protein COH2::FLAG in 2 isolates of the strain *WT-COH2-FLAG* and 5 isolates of the strain *Coh1$^{OE}$-COH2-FLAG*. (B) Co-IP analysis shows that the COH2 protein dimerized in the *Coh1*-overexpressing strain *Coh1$^{OE}$*. Three strains were used: (1) *WT-COH2-Myc*, constitutively expressing the fusion protein COH2::Myc; (2) *WT-COH2-FLAG/COH2-Myc*, constitutively expressing the fusion proteins COH2::FLAG and COH2::Myc; and (3) *Coh1$^{OE}$-COH2-FLAG/COH2-Myc*, which constitutively expressed the COH1 protein and the fusion proteins COH2::FLAG (molecular weight = 33 kDa) and COH2::Myc (molecular weight = 44.6 kDa). The strains were grown in the nutrient-rich medium SDY, where COH1 was not expressed in the strain *WT-COH2-FLAG/COH2-Myc*. Immunoprecipitations were conducted with an anti-FLAG antibody. Proteins were detected by immunoblot analysis with anti-Myc or anti-FLAG antibodies. Note: The fusion proteins COH2::FLAG and COH2::Myc dimerized in the strains *Coh1$^{OE}$-COH2-FLAG/COH2-Myc* and *WT-COH2-FLAG/COH2-Myc*, and the fusion protein COH2::Myc formed a dimer in the strain *WT-COH2-Myc*. (C) GFP-tagged N-terminus of COH2 enters into the nucleus during saprophytic growth in SDY medium (left panel) and real hemocoel colonization (right panel). Two strains were used: (1) *WT-COH2-N-GFP*, with the fusion protein COH2-N::GFP (GFP fused to the COH2

N-terminus) expressed in the WT strain, and (2) *Coh1^OE^-COH2-N-GFP*, constitutively expressing the COH1 protein and COH2-N::GFP. Three independent isolates of the strain *Coh1^OE^-COH2-N-GFP* were randomly selected for this assay. In each panel: left, bright field microscopy; middle, fluorescence microscopy for DAPI (4′,6-diamidino-2-phenylindole) staining; right, fluorescence microscopy for GFP observation. C, conidium; N, nucleus. Note: In all tested strains, the GFP fluorescence intensity was strongest in the nucleus. (D) Co-IP analysis shows that the GFP-tagged N-terminus of COH2 (COH2-N::GFP) could not physically contact the HA-tagged COH1 (COH1::HA). Immunoprecipitation was conducted with anti-HA antibody. Proteins were detected by immunoblot analysis with anti-HA or anti-GFP antibodies. The data underlying all the graphs shown in this figure can be found in S1 Data. (TIF)

**S9 Fig. *Coh2* is involved in infection of insects, but not in saprophytic growth.** (A) qRT-PCR analysis of *Coh2* expression during surrogate hemocoel colonization (Hemolymph), cuticle penetration (Cuticle), and saprophytic growth (SDY). (B) GFP signal in fungal cells of the strain with *gfp* driven by *Coh2* promoter on the cuticle (Cuticle) and in the real hemocoel (Hemocoel) of *G. mellonella* larvae. F, fungal cells. Scale bar: 10 μm. Images are representative of 3 independent experiments. (C) Colony diameter on PDA plates of the WT strain, the mutant *ΔCoh2*, and its complemented strain *C-ΔCoh2*. Inoculations were conducted by applying 5 μL of a conidial suspension ($1 \times 10^7$ conidia/mL) on the center of a PDA plate. Note: Within each day, no significant difference in colony diameter was observed between the 3 strains. (D) Conidial yields. Note: No significant difference was seen between the 3 strains. The insets are the colony pictures that were taken at day 18 after inoculation on PDA plates (scale bar represents 10 mm). (E) Appressorial development on a hydrophobic plastic surface. At each time point, values with different letters are significantly different ($n = 3$, $P < 0.05$, Tukey's test in one-way ANOVA). (F) Cuticle penetration. This experiment is described in S2I Fig. The scale bar represents 1 cm. Images are representative of 3 independent experiments. (G and H) $LT_{50}$ values via (G) topical application or (H) injection. Data are expressed as mean ± SE. Values with different letters are significantly different ($P < 0.05$, Tukey's test in one-way ANOVA). The data underlying all the graphs shown in this figure can be found in S1 Data. (TIF)

**S10 Fig. Identification of the DNA motif bound by COH2.** (A) Distribution of all COH2 peak locations within the genome revealed by ChIP-Seq analysis. (B) The top 3 binding motifs enriched by the HOMER (Hypergeometric Optimization of Motif EnRichment) software. (C) SDS-PAGE analysis of the expression and purification of the recombinant COH2-DBD (DNA binding domain of COH2) protein in *E. coli*. M, protein ladder (Thermo Fisher Scientific). Lane 1: Crude extract from the *E. coli* cells with the empty plasmid pET-28a (control). Lane 2: Crude extract from the *E. coli* cells expressing COH2-DBD. Lane 3: The supernatant of the crude extract shown in lane 2. Lane 4: Proteins purified from the supernatant shown in lane 3 with the HisPur Ni-NTA Resin. (D) Western blot analysis to confirm the expression of the recombinant COH2-DBD using the anti-His tag antibody. M, protein ladder. Lanes 1 and 2 are the proteins shown in lanes 1 and 4, respectively, in (C). (E) The sequences of the DNA probes containing the motif *COH2-BM* (shown in red), which is flanked by the sequences (black) in the promoter of the gene MAA_04430. The name of the DNA probe is shown on the left, and its sequence is on the right. Wild type: the DNA probe containing the wild-type motif *COH2-BM*; 1A: the nucleotide at position 1 in the motif *COH2-BM* is changed from the wild-type one (T) into the nucleotide A. The naming system is also used for all other mutated DNA probes shown in this figure. (F) EMSA analysis of the binding of the DNA probes to the

recombinant COH2-DBD protein. The binding activity was demonstrated by the DNA (Bio-probe: the biotin-labeled wild-type DNA probe) band shift prior to the addition of the specific competitor (Cold probe: the unlabeled wild-type DNA probe) in a 300-fold excess. The importance of each nucleotide in the motif *COH2-BM* in its binding to the recombinant COH2-DBD protein is shown by the impact on the DNA (Bio-probe) band shift of adding an unlabeled mutated probe (Mut.Cold-probe) as a competitor in a 300-fold excess. The names of the unlabeled mutated DNA probes are shown on the top of the lanes. The images are a representative of 3 independent experiments.
(TIF)

**S11 Fig. Analysis of genes regulated by COH1 and COH2 during hemocoel colonization or cuticle penetration.** (A) A summary of differentially expressed genes profiled by RNA-Seq analysis between the WT strain and the mutants ΔCoh1 and ΔCoh2 during surrogate hemocoel colonization (Hemolymph) or cuticle penetration (Cuticle). Up (red) and down (green) represent the genes up-regulated and down-regulated in the mutants compared with the WT strain. (B) Venn diagram showing the distribution of genes positively regulated by COH1 and negatively regulated by COH2 during surrogate hemocoel colonization. (C) qRT-PCR analysis of *Hat1* expression during surrogate hemocoel colonization in the WT strain, the mutants ΔCoh1 and ΔCoh2, the *Coh1*-overexpressing strain *Coh1*[OE], and the *Coh2*-overexpressing strain *WT-COH2-FLAG*. (D) qRT-PCR analysis of expression of the 4 destruxin biosynthesis genes (*DtxS1*, *DtxS2*, *DtxS3*, and *DtxS4*) during surrogate hemocoel colonization in the WT strain and the mutants ΔCoh1 and ΔCoh2. (E) GFP signal in the fungal cells collected from the real hemocoel of insects infected by the strains *WT-PDtxS3-GFP*, *ΔCoh1-PDtxS3-GFP*, and *ΔCoh2-PDtxS3-GFP*. In the 3 strains, the *gfp* gene was driven by the *DtxS3* promoter. F, fungal cells; H, hemocyte; MGV, mean gray value. Scale bar represents 10 μm. (F) RNA-Seq analysis of regulation of the chitinase, protease, lipase, and P450 genes by COH2 during cuticle penetration. The expression level of a gene in the WT strain is set to 1; the values represent the log2-transformed fold-changes of differential gene expression in the mutant ΔCoh2. The 3 individual experiments are indicated by the numbers 1, 2, and 3, respectively. (G) qRT-PCR analysis of expression of cuticle-degrading genes during cuticle penetration in the mutant ΔCoh2 and the WT strain. For quantification of free amino acids, peptides, and activities of cuticle-degrading enzymes, the fungal strains were grown in cuticle medium using *G. mellonella* cuticle as the sole carbon and nitrogen source. (H) Total extracellular protease activity in culture supernatants. (I and J) The concentrations of (I) free amino acids and (J) peptides in culture supernatants. The control was insect cuticle medium that was not inoculated with fungi. (K) The chitinase activity in culture supernatants. All assays were repeated 3 times. Values with different letters are significantly different ($P < 0.05$). C-ΔCoh2: the complemented strain of ΔCoh2. The data underlying all the graphs shown in this figure can be found in S1 Data.
(TIF)

**S1 Raw Images. Original Western blot and gel images.**
(PDF)

**S1 Table. Primers used in this study.**
(DOCX)

## Acknowledgments

We thank Dr. Jun Chen at the Institute of Ecology at Zhejiang University for his help with retrieval of the promoter sequences from *M. robertsii* genome.

## Author Contributions

**Data curation:** Xing Zhang.

**Formal analysis:** Xing Zhang.

**Funding acquisition:** Weiguo Fang.

**Investigation:** Xing Zhang.

**Methodology:** Yamin Meng, Yizhou Huang, Dan Zhang.

**Project administration:** Weiguo Fang.

**Supervision:** Weiguo Fang.

**Writing – original draft:** Xing Zhang.

**Writing – review & editing:** Weiguo Fang.

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
