## [Editor Report · Decision Letter 0]

25 Aug 2020

Dear Dr Zhang, 

Thank you for submitting your manuscript entitled "Unveiling a novel cascade regulating the response of Metarhizium robertsii to distinct microenvironments during infection of insects" for consideration as a Research Article by PLOS Biology.

Your manuscript has now been evaluated by the PLOS Biology editorial staff as well as by an academic editor with relevant expertise and I am writing to let you know that we would like to send your submission out for external peer review.

Please re-submit your manuscript within two working days, i.e. by Aug 27 2020 11:59PM.

Kind regards,

Ines

--

Ines Alvarez-Garcia, PhD

Senior Editor

PLOS Biology

Carlyle House, Carlyle Road

Cambridge, CB4 3DN

+44 1223–442810

---

## [Decision Letter · Decision Letter 1]

30 Oct 2020

Dear Dr Zhang,

Thank you very much for submitting your manuscript "Unveiling a novel cascade regulating the response of Metarhizium robertsii to distinct microenvironments during infection of insects" for consideration as a Research Article at PLOS Biology. Thank you also for your patience as we completed our editorial process, and please accept sincere my apologies for the delay in providing you with our decision. Your manuscript has been evaluated by the PLOS Biology editors, an Academic Editor with relevant expertise, and by three independent reviewers.

The reviews of your manuscript are appended below. You will see that the reviewers find your work potentially interesting, however they also raise several concerns that would need to be address if we decide to pursue the manuscript for publication. Both Reviewers 1 and 2 have relatively minor points, but Reviewer 3 is more critical and thinks you cannot formally exclude that you are monitoring a differential response to two different insect species, locust in the case of cuticle and moth for the hemolymph. After consulting with the academic editor, we deem this point essential for publication and you will need to provide a convincing response to address this issue. Ideally if you want to study differences to different tissues it would have to be in the same host. In addition, this reviewer thinks you should validate the model in real infection conditions and use antibodies against COH2 to show that the expression of Coh1 leads to the degradation of COH2 during hemolymph colonisation. This reviewer also suggests other experiments to demonstrate that the Coh1 phenotype is not due to a secondary mutation, which would be essential to address and requires either an independent mutant or genetic complementation, and the generation of a Coh1-Coh2 double mutant, which we don’t consider a requirement for publication. The academic editor has also mentioned that the quality of the figures should be improved, and notes an excessive use of histograms over scatterplots and poorly controlled experiments such as the coIP in Figure 3B.

Based on these points, I regret that we cannot accept the current version of the manuscript for publication. However we remain interested in your study and we would be willing to consider resubmission of a comprehensively revised version that thoroughly addresses all the reviewers' comments. We cannot make any decision about publication until we have seen the revised manuscript and your response to the reviewers' comments. Your revised manuscript would be sent for further evaluation by the reviewers.

We appreciate that these requests represent a great deal of extra work, and we are willing to relax our standard revision time to allow you six months to revise your manuscript.We expect to receive your revised manuscript within 6 months.

**IMPORTANT - SUBMITTING YOUR REVISION**

*Resubmission Checklist*

*Published Peer Review*

*PLOS Data Policy*

*Blot and Gel Data Policy*

Sincerely,

Ines

--

Ines Alvarez-Garcia, PhD,

Senior Editor,

ialvarez-garcia@plos.org,

PLOS Biology

Reviewers’ comments

Rev. 1:

Cuticular penetration and hemocoel colonization are cellular processes and events critical for the pathogenesis of a fungal insect pathogen. This interesting manuscript presents massive data from technically robust experiment performed to characterize a novel cascade that regulates the pathogenesis of Metarhizium robertsii. The cascade comprises a pair of core proteins, namely the regulatory protein COH1 expressed exclusively in hemocoel colonization and the transcription factor COH2 localized in nucleus. The COH1-coding gene was evidently required for the fungal virulence via normal cuticle infection and transcriptionally mediated by HADC1 and HAT1, two key enzymes regulating histone H3 acetylation in the promoter region of the gene. COH1 was found interacting with the N-terminus of COH2 but not affecting an entry of COH2 into the nucleus. The interaction can reduce COH2 stability through a proteasome-dependent pathway and derepress the COH2-mediated repression of genes involved in hemocoel colonization. COH2 activates the expression of cuticle degradation-related genes during normal cuticle infection by directly binding to their promoter regions but those genes are repressed by COH1 during hemocoel colonization. These findings unveil a novel mechanism underlying the fungal adaptation to the microenvironments insect cuticle and hemocoel encountered during the normal infection and after cuticular penetration respectively and offer an in-depth insight into the impressive roles of COH1 and COH2 in coordinating the processes of cuticle infection and intrahemocoel propagation post-infection

Overall, all data are well organized and presented. The manuscript needs a careful revision to improve clarity and conciseness.

Labels and legends in most figures are too busy or not proportional to chart size. Many bar charts can be improved by proper adjustments of bar-to-bar spaces, labels and legends. Using log10 scale in some bar charts may help to show big differences of gene transcripts between strains.

Rev. 2:

The manuscript entitled "Unveiling a novel cascade regulating the response of Metarhizium robertsii to distinct microenvironments during infection of insects" reports the functional characterization of two interacting transcription factors (COH1 and COH2) regulating gene expression of a parasitic fungus on the cuticle or within the hemocoel of the infected host insect. COH1 was expressed upon epigenetic repression conferred by the histone deacetylase and the histone acetyltransferase. The study is interesting and the authors have used a considerable spectrum of methods to elaborate a convincing data set. I appreciate the extended spectrum of experimental approaches such as yeast-two-hybrid system to determine interactions between COH1 and COH2. I have some minor comments to improve the manuscript constructively:

In the introduction, the authors should mention the important role of proteolytic enzymes during penetration of the cuticle. Some of these enzymes including metalloproteases have been recognized to operate as virulence factors whose control is important.

The authors should discuss what factors could trigger the expression of COH1 in the hemocoel.

The entity of recent work implicates that transcriptional reprogramming in M. robertsii and other entomopathogenic fungi during infection is more plastic than previously thought and that they respond in a specific manner to host defense molecules.

Rev. 3:

In this study, Weiguo Fang and coworkers focus on a gene, Coh1, found to be expressed in conditions thought to reproduce the ones happening during the colonization of insect hemocoel by the entomopathogenic fungus Metarhizium robertsii, that is, growth in collected cell-free Galleria mellonella hemolymph. They check by RTqPCR that it is not expressed when the fungus is grown in SDY medium nor in medium containing locust cuticle as the sole carbon and nitrogen source. They link this specific pattern of expression to a modification of epigenetic marks that would result from the decreased expressions of the Hat1 histone acetyl transferase and Hdac1 histone deacetylase genes at this specific stage. The involvement of enzymes with two opposing activities would reflect the positive action of HAT1 on Hdac1 transcription, thereby resulting in the expression of Coh1 only when the conidia are incubated with hemolymph. Coh1 deletion mutants display a strongly decreased virulence when topically applied to G. mellonella larvae. The direct injection of conidia led to a much milder phenotype of decreased virulence. Of note, the three complementation constructs expressing the wild-type Coh1 gene from distinct genomic locations (random integration?) failed to rescue the mutant phenotype, possibly because of the different chromatin environment in which the wild-type gene is inserted in each transformant. The authors next use a strategy that makes little common sense to identify a bZIP transcription factor, COH2, that happens to interact with COH1. They show that the binding of COH1 to COH2 destabilizes the transcription factor that likely gets degraded through the proteasome. The authors next show that COH2 is involved in pathogenicity, the corresponding gene being required for piercing through the insect cuticle. It is thus difficult to establish whether this gene also plays a role in the phase of hemocoel invasion, as it can only be addressed by injecting conidia; they find a decreased virulence phenotype that is as weak as that displayed by Coh1 mutants, even though it was not expected to as COH2 is supposedly unstable during this stage of the infection. They characterize in detail the DNA binding motif of COH2 and use this knowledge to interpret RNAseq experiments performed during "hemocoel colonization" by the two mutants vs. wild-type. They conclude that Coh1 is required to repress the expression of some Coh2 regulated genes involved in piercing the cuticle through its enzymatic degradation as well as to stimulate the expression of genes encoding toxins that are presumably required for successful invasion of the host.

1. While this study is rather strong on the molecular biology of the Coh1 and Coh2 genes, it sorely lacks a validation of the proposed model under real infection conditions. The authors over-rely on the postulate that the artificial conditions they use accurately reproduce the actual situation occurring in vivo during infection. This is already reflected in the title and throughout the manuscript when for instance employing the term "hemocoel colonization", a clear overstatement. For instance, the authors cannot formally exclude that what they are monitoring is a differential response to two different insect species, locust in the case of cuticle and moth for the hemolymph. More importantly, the hemolymph is collected from naive larvae and lacks immune cells: it cannot be excluded that the expression of genes if the fungus is altered by the exposure host defenses. While this reviewer understands the practical issues that lead to use this set-up, it is absolutely indispensable to validate the findings in vivo at different stages of the infection. They thus have to monitor the steady-state levels of Coh1, Coh2, Hat1, Hdac1, and representative genes regulated by Coh1 or Coh2 under in vivo conditions. Of note, they likely could retrieve hyphal bodies from the hemolymph of infected larvae. The current title is also inaccurate as long as these points have not been addressed.

2. The general model is that the expression of Coh1 leads to the degradation of COH2 during "hemolymph colonization". It would be ideal to demonstrate this in vivo using an antibody raised against COH2. At the very, least, the authors could check the tagged COH2 construct comparing wild-type vs. Coh1 deletion mutants after infection. What happens to cuticle penetration and virulence in the topical infection model when Coh1 is overexpressed? Conversely, what happens when Coh2 overexpressing conidia (in wild-type or Coh1 mutant background) are injected into G. mellonella larvae? Is there indeed a stronger melanization reaction and are antimicrobial peptide genes induced more strongly? The generation of a Coh1-Coh2 double mutant and its phenotypic analysis after injection would allow determining whether the Coh1 phenotype is only mediated through COH2 degradation. Of note, the fungal burden should be measured in all in vivo experiments just prior to the onset of larval death, for both infection models.

3. The authors provide an interesting explanation for the lack of rescue of the Coh1 mutant phenotypes by the expression of the wild-type gene. However, they still cannot formally exclude that the Coh1 phenotype is due to a second-site mutation. The easiest to exclude this possibility would be to use the Coh1 overexpression construct using the pTef promoter. The alternative would be to knock-in back the wild-type gene in the deletion mutant. With respect to the hypothesis of the Hat1 action being mediated through repression of Hdac1 expression, the authors should establish whether Hdac overexpression from the pTef promoter overrules the Hat1 deficiency phenotype as regards Coh1 expression.

4. One important question remaining is how the expression of Hat1 gets decreased under conditions of growth in naive cell-free hemolymph. The authors should indicate whether the expression of this gene is affected when Coh1 or Coh2 are overexpressed or in the deletion mutant lines.

5. The authors rely on constructs leading to the expression of tagged proteins. Their conclusions would be much stronger if they were to demonstrate that the resulting tagged proteins are functional, that is, that they can complement the corresponding deletion mutant phenotype (of course, this does not apply to constructs expressing truncated proteins.

6. Could COH1 be linked to ubiquitinylation processes (not to be addressed experimentally in this study)?

Minor comments:

a) Line 32 : COH2 not COH

b) Line 65 : is Ref. 8 that relevant here ?

c) Lines 63-69 are taken with some modifications from one of their previous publications (ref. 10), a practice the authors may want to avoid.

d) Line 116 and others: it is irritating to have figure legends in the middle of the text.

e) Lines 135-144: does the Coh1 sequence encode a signal peptide (presumably not), a NLS?

f) Lines 270-271: the logic of the experiment remains obscure and citing a single (lonely?) study does not help understanding it.

g) Line 404: the COH2-BM sequence given here does not correspond to the consensus nor to the sequence shown in figures. This reviewer did not really observe an inverse repeat in the consensus motif; would a single nucleotide spacer be enough for the binding motif of a dimer, e.g., would the two "repeats" be on the same side of the double-helix?

h) Fig. 2B, D: Are the SDY and hemolymph conditions each normalized to the wild-type? Would it not be better to normalize the expression to a single condition, e.g., SDY wild-type, so that the expression in SDY and hemolymph can really be compared?

---

## [Decision Letter · Decision Letter 2]

28 Apr 2021

Dear Dr Zhang,

Thank you very much for submitting a revised version of your manuscript "Unveiling a novel cascade regulating the response of Metarhizium robertsii to distinct microenvironments during infection of insects" for consideration as a Research Article at PLOS Biology. This revised version of your manuscript has been evaluated by the PLOS Biology editors, the Academic Editor and one of the original reviewers.

In light of the reviews (attached below), we are pleased to offer you the opportunity to address all the remaining points raised by the reviewer in a revised version that we anticipate should not take you very long. We will then assess your revised manuscript and your response to the reviewer' comments and we may consult the reviewers again.

We expect to receive your revised manuscript within 1 month.

**IMPORTANT - SUBMITTING YOUR REVISION**

3. Resubmission Checklist

a) *Published Peer Review*

b) *PLOS Data Policy*

Thank you for sending the S1_Data file containing the data underlying all the graphs shown in the figures. We have spotted some mistakes in the Fig. S11 tab that should be amended: please add Fig. S11A data, as it seems to be missing, and relabel the data currently shown as S11B Fig – it should be S11C Fig.

Please also make sure you mention in the corresponding figure legends WHERE THE DATA CAN BE FOUND.

Please ensure that your Data Statement in the submission system accurately describes where your data can be found and also please make publicly available the RNA-Seq data that you have deposited in GenBank with accession number PRJNA637940.

c) *Blurb*

Please also provide a blurb which (if accepted) will be included in our weekly and monthly Electronic Table of Contents, sent out to readers of PLOS Biology, and may be used to promote your article in social media. The blurb should be about 30-40 words long and is subject to editorial changes. It should, without exaggeration, entice people to read your manuscript. It should not be redundant with the title and should not contain acronyms or abbreviations. For examples, view our author guidelines: https://journals.plos.org/plosbiology/s/revising-your-manuscript#loc-blurb

Sincerely,

Ines

--

Ines Alvarez-Garcia, PhD,

Senior Editor,

PLOS Biology

Reviewers' comments

Rev. 3:

The authors have to be lauded for their efforts to address the issues raised by reviewers. 

A critical point is that of the validation of their model in vivo during infection. To this end, the authors have used several tagged constructs to monitor whether the genes under investigation are indeed expressed in vivo. They rely on fluorescence to assess reporter expression. While in some cases such as Fig. 1E, the results are clear-cut, they are less so in other instances such as Fig. 3D and especially Fig. 5G and Fig. S11E. The fluorescence should be measured on several microscopic fields and rigorously quantified, subtracting the general background of the picture.

The results displayed in Fig. S8C showing a nuclear localization of COH2 occurring even when Coh-1 is overexpressed is puzzling since the model of the authors is that COH1 gets expressed only when the fungus is in the hemolymph and leads to the degradation of COH2. In addition, since COH1 lacks a nuclear localization signal, it may not reside in the same subcellular compartment, raising the possibility that COH2 in the nucleus might be "immune" from the action of COH1 in the cytoplasm. Presumably, these data have been gained with the fungus grown on SDY medium (the conditions are not described in the text nor in the figure legend). Would it be possible to use this strain in infections, even though the functionality of the N-GFP-COH2 fusion has not been tested? Can the authors exclude the possibility that adding GFP to the N-terminal side prevents COH1 from binding to COH2?

The authors should also mention in the Discussion another consequence from the analysis of the double Coh1-Coh2 mutant. One might have reasoned that the Coh1 deletion phenotype might be solely caused by the ectopic stabilization of COH2. That the double mutant is still less virulent strongly suggests that COH1 is regulating other target proteins, in line with the results of the RNAseq analysis. 

The clarity of the manuscript could be substantially improved by clearly stating the conditions used: the authors repeatedly refer to "hemocoel colonization" when they are using the assay of incubating the fungus in collected hemolymph that has been diluted with an anticoagulant solution (lines, 103, 120, 137, 225, 227, 248, 268, 272, 289, 385, 463, 468, 488, 506, 556, 559, 560, 620?, 631, 637, 788950, 961, 1035, 1425 and likely others I missed. It is important to use a term such as "surrogate hemocoel colonization" to make the distinction between artificial conditions and the in vivo situation. The use of the term "real hemocoel" when referring to in vivo conditions does not suffice. By the same token, the term "cuticle penetration" should likewise be avoided (lines 532, 554, 558).

Minor points: 

 a) it might be desirable to use scatter plots when displaying the results of gene expression measured by RTqPCR. 

 b) Line 318: please, change "COH1 physically interacted with" to "COH1 can physically interact with": these are artificial conditions in which the proteins are overexpressed. Whether the two proteins interact in vivo remains a possibility, but see also above.

 c) Fig. S2F-G: the Y-axis caption should be changed as qPCR was used.

---

## [Editor Report · Decision Letter 3]

18 Jun 2021

Dear Dr Zhang,

Thank you for submitting your revised Research Article entitled "Unveiling a novel cascade regulating the response of Metarhizium robertsii to distinct microenvironments during infection of insects" for publication in PLOS Biology. I have now obtained advice from the original Academic Editor and discussed the revision with the editorial team. 

Based on this advice, we will probably accept this manuscript for publication. While we are now mostly satisfied with the changes you have done in the revision, we think that the quality of the IPs shown in Fig. S8D are poor. It would be useful if these could be improved if possible and you should temper your statement regarding the lack of interaction given that this is only supported by the coIP and that positive controls to show that COH1::HA remains intact in the IP are missing.

Please also make sure to address the following data-related request:

- Many thanks for amending the data file to include the correct data for Figure S11 and to mention in the figure legends where the data can be found. It seems you have not mentioned where the data can be found in the figure legend of Fig. S1, thus please add this information.

In addition, we would like you to consider a suggestion to improve the title:

"A novel cascade allows Metarhizium robertsii to distinguish cuticle and hemocoel microenvironments during infection of insects"

We expect to receive your revised manuscript within two weeks. 

*Published Peer Review History*

*Early Version*

Sincerely,

Ines

--

Ines Alvarez-Garcia, PhD,

Senior Editor,

ialvarez-garcia@plos.org,

PLOS Biology

---

## [Editor Report · Decision Letter 4]

9 Jul 2021

Dear Dr Zhang,

On behalf of my colleagues and the Academic Editor, Sophien Kamoun, I am pleased to say that we can in principle offer to publish your Research Article "A novel cascade allows Metarhizium robertsii to distinguish cuticle and hemocoel microenvironments during infection of insects" in PLOS Biology, provided you address any remaining formatting and reporting issues. These will be detailed in an email that will follow this letter and that you will usually receive within 2-3 business days, during which time no action is required from you. Please note that we will not be able to formally accept your manuscript and schedule it for publication until you have made the required changes.

PRESS

Sincerely, 

Ines

--

Ines Alvarez-Garcia, PhD 

Senior Editor 

PLOS Biology
